# An evidence based hypothesis on the existence of two pathways of mitochondrial crista formation

**Max E Harner[1,2], Ann-Katrin Unger[3], Willie JC Geerts[4], Muriel Mari[5], Toshiaki Izawa[1], Maria Stenger[3], Stefan Geimer[3], Fulvio Reggiori[5], Benedikt Westermann[3], Walter Neupert[1,6]***

[1]Max Planck Institute of Biochemistry, Martinsried, Germany; [2]Walter Brendel Centre of Experimental Medicine, Ludwig-Maximilians-Universität München, Martinsried, Germany; [3]Cell Biology and Electron Microscopy, Universität Bayreuth, Bayreuth, Germany; [4]Biomolecular Imaging, Bijvoet Center, Universiteit Utrecht, Utrecht, Netherlands; [5]Department of Cell Biology, University Medical Center Groningen, University of Groningen, Groningen, Netherlands; [6]Department of Anatomy and Cell Biology, Biomedical Center, Ludwig-Maximilians Universität München, Martinsried, Germany

**Abstract** Metabolic function and architecture of mitochondria are intimately linked. More than 60 years ago, cristae were discovered as characteristic elements of mitochondria that harbor the protein complexes of oxidative phosphorylation, but how cristae are formed, remained an open question. Here we present experimental results obtained with yeast that support a novel hypothesis on the existence of two molecular pathways that lead to the generation of lamellar and tubular cristae. Formation of lamellar cristae depends on the mitochondrial fusion machinery through a pathway that is required also for homeostasis of mitochondria and mitochondrial DNA. Tubular cristae are formed via invaginations of the inner boundary membrane by a pathway independent of the fusion machinery. Dimerization of the $F_1F_O$-ATP synthase and the presence of the MICOS complex are necessary for both pathways. The proposed hypothesis is suggested to apply also to higher eukaryotes, since the key components are conserved in structure and function throughout evolution.

*For correspondence: Neupert@biochem.mpg.de

**Competing interests:** The authors declare that no competing interests exist.

## Introduction

Mitochondria have a multitude of functions in the eukaryotic cell. They perform respiration-dependent energy transduction to generate proton-motive force and ATP, and house a large number of metabolic enzymes. They have their own genetic system but are closely connected with the rest of the cell by a number of pathways, such as metabolite, protein and lipid transport, protein quality control, autophagy and apoptosis. Light microscopy has revealed mitochondria as threadlike bodies that form three-dimensional networks. Mitochondria are dynamic organelles undergoing continual fission and fusion (*Bereiter-Hahn and Vöth, 1994*). The dynamics of mitochondria is essential for their function as well as maintenance, inheritance and integrity of their DNA (mtDNA). Recent research has revealed the involvement of this organelle in a multitude of human pathological conditions, including neurodegeneration, cardiomyopathies, metabolic diseases and cancer (*Chan, 2012*; *Costa and Scorrano, 2012*; *Itoh et al., 2013*; *Mourier et al., 2014*; *Nunnari and Suomalainen, 2012*; *Pickrell and Youle, 2013*; *Youle and van der Bliek, 2012*).

**eLife digest** Cells contain compartments called mitochondria, which are often called the powerhouses of the cell because they provide energy that drives vital cellular processes. Mitochondria have two membranes: an outer and an inner membrane. The outer membrane separates the mitochondria from the rest of the cell. The inner membrane is elaborately folded and the folds – called cristae – create a larger space to accommodate all of the protein machinery involved in producing energy. The cristae can be shaped as flat sac-like structures called lamellar cristae or as tubes known as tubular cristae.

Mitochondria are dynamic and are constantly fusing with other mitochondria and splitting up. Even though the internal architecture of mitochondria was first revealed around 60 years ago, it is still not clear how the cristae form. Harner et al. now address this question in yeast cells by combining imaging, biochemistry and genetic approaches.

The experiments show that lamellar cristae form when two mitochondria fuse with each other. The outer membranes merge and then the inner membranes start to fuse around their edges to generate the sac-like structure of lamellar cristae. A yeast protein called Mgm1 (known as Opa1 in mammals) drives the fusion of the inner membranes, but this process only takes place when enzymes called $F_1F_O$-ATP synthases on the inner membrane form pairs with one another. These $F_1F_O$-ATP synthase pairs stabilize the cristae membranes as they curve to form the sac-like structure. Later on, the formation of a group of proteins called the MICOS complex halts the fusion process to prevent the lamellar cristae from completely separating from the rest of the inner membrane.

Harner et al. also found that tubular cristae form using a different mechanism when the inner membrane of the mitochondria grows inwards. This process also requires pairs of $F_1F_O$-ATP synthases and the MICOS complex, but does not involve Mgm1/Opa1. Together, these findings show that lamellar and tubular cristae in yeast form using two different mechanisms. Since the key components of these mechanisms are also found in virtually all other eukaryotes, the findings of Harner et al. are also likely to apply to many other organisms including animals.

Mitochondrial ultrastructure, also referred to as the mitochondrial architecture, is intimately linked to the function and homeostasis of this organelle, and has therefore been investigated in a large number of cell types, tissues and organisms (*Fawcett, 1981*). There is a considerable diversity in mitochondrial architecture, but common basic structural elements can be recognized (*Mannella, 2008*; *Perkins et al., 2010*; *Sun et al., 2007*). Mitochondria are delimited from the cytosol by the outer membrane (OM). The inner membrane (IM) is composed of the inner boundary membrane (IBM) and the crista membranes. The cristae are invaginations of the IM into the interior of the mitochondria (*Figure 1*). In most cases they form lamellae, but tubules are also observed (*Fawcett, 1981*). Crista junctions (CJs) join the cristae with the IBM. They represent small ring or slot like openings that connect the intermembrane space with the intracrista space. The IM encloses the matrix space. Cristae do not cross the matrix space completely, but they are closed by strongly bent crista rims. Septa completely cross the matrix space and separate it into distinct compartments. They are virtually absent in wild type (WT) cells, but are frequent in a variety of mutant cells. Continuous protein and lipid import leads to generation of curved septa which can be seen as onion-like structures in cross sections (*Figure 1*). Over the past 20 years a number of proteins have been identified which play key roles in mitochondrial function, dynamics and homeostasis. Still, our knowledge concerning the molecular mechanisms that determine the formation and maintenance of mitochondrial architecture is quite limited.

A molecular understanding of these complex processes is possible only when biochemical and genetic analyses are combined with 3D electron microscopy (EM) analysis, whose resolution is sufficient to precisely evaluate the ultrastructure of mitochondrial membranes and localize proteins with nano-scale precision. With this as a guiding principle, we have addressed a central question of the molecular biology of mitochondria, the mechanisms of the biogenesis and maintenance of cristae. This question is open ever since the structural organization of mitochondria was discovered (*Palade, 1953*; *Sjostrand and Hanzon, 1954*).

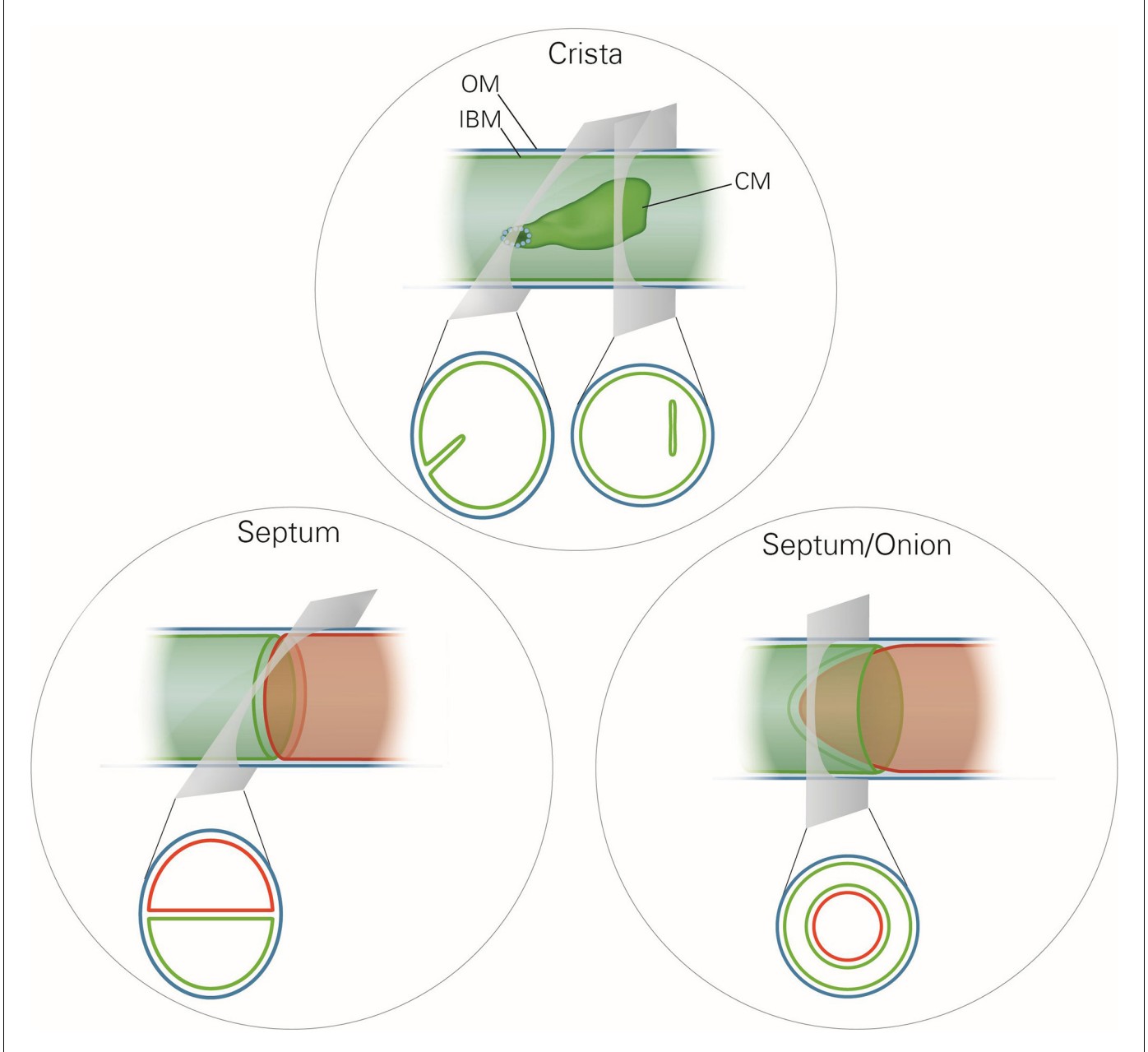

**Figure 1.** Schematic representation of the ultrastructure of the mitochondria. Upper panel, a wild type mitochondrion containing a crista. Lower left panel, a mitochondrion containing a septum. A septum separates the matrix into two discontinuous subcompartments. Lower right panel, a mitochondrion containing an extended curved septum, appearing as an onion-like profile in EM. The drawings illustrate the reconstruction of the 3D structures from the 2D EM sections whose directions are indicated by the grey planes. Blue line, outer membrane (OM); green line, inner membrane (inner boundary membrane (IBM), crista membrane (CM) and septa membranes); small light rings, assembled MICOS complex forming a crista junction. In the lower panels, the inner boundary membranes of two matrix subcompartments are drawn in red and green to point out that the matrix is divided by a septum.

Our approach has led us to an entirely novel hypothesis on the formation and homeostasis of cristae and CJs. Based on our results we present a model that postulates the existence of two different pathways of crista formation and is fully consistent with published data on mitochondrial structure, function and dynamics in yeast and mammalian cells. In the first pathway, two IBM sheets are converted into a single lamellar crista upon fusion of mitochondria. This process not only requires

Mgm1, the dynamin-like fusion protein of the IM, but also the dimeric form of the $F_1F_O$-ATP synthase ($F_1F_O$) and the MICOS complex. Mgm1 appears to mediate fusion of the IM only when dimeric $F_1F_O$ is available to stabilize bending of the newly formed crista membrane and to thereby generate crista rims. Assembly of the MICOS complex is proposed to limit the fusion process by forming a CJ. In the second pathway, crista formation is independent of Mgm1, but dependent on dimeric $F_1F_O$ and MICOS. It entails formation of tubular cristae and not of lamellar cristae. Our results suggest a mechanism by which growth of membranes from the IBM, driven by the influx of newly synthesized proteins and lipids, creates these tubular cristae. Importantly, all proteins found to be involved in crista biogenesis are conserved from yeast to human, emphasizing the general importance of this process.

## Results

### Mgm1 plays a direct role in cristae formation

The dynamin-related GTPase, Mgm1 (yeast)/Opa1 (higher eukaryotes), is essential for fusion of the mitochondrial IM. Its deletion in yeast leads to fragmentation of mitochondria and loss of respiration-dependent growth as well as of mtDNA (*Cipolat et al., 2004*; *Jones and Fangman, 1992*; *Meeusen et al., 2006*; *Song et al., 2009*; *Wong et al., 2000*). Previous EM analyses showed an altered IM structure in Δ*mgm1* cells that includes the loss of cristae (*Sesaki et al., 2003*). Consistent with these observations, quantitative EM of Δ*mgm1* cells revealed mitochondrial profiles that were mainly empty or contained one or a few septa. Vesicular and crista-like membranes were present only to a minor extent (*Figure 2—figure supplement 1*). Moreover, the levels of mitochondrial respiratory components were strongly reduced (*Figure 2—figure supplement 2*).

These observations raised the possibility that Mgm1 is required for the formation of cristae. Cristae membranes accommodate the respiratory chain complexes which consist of both nuclear and mitochondria-encoded subunits. Thus, it is conceivable that loss of mtDNA first leads to the loss of respiratory chain complexes and then indirectly also to the loss of cristae. Alternatively, Mgm1 might be required for cristae formation, and in the absence of cristae mtDNA is not maintained. To discriminate between these two scenarios, we made use of the temperature sensitive *mgm1-5* mutant in which a shift to non-permissive temperature leads to the inactivation of the protein and concomitant fragmentation and alteration of mitochondrial ultrastructure (*Meeusen et al., 2006*; *Wong et al., 2000*). We performed quantitative EM of WT and *mgm1-5* cells grown at 25°C, shifted to 37°C for 25 min, and back again to 25°C for 30 min. In WT cells almost only cristae were present and no significant changes were observed upon exposure to 37°C and return to 25°C (*Figure 2A and B*). In *mgm1-5* cells grown at 25°C, cristae made up about 70%; apparently the temperature sensitive mutant was leaky. Exposure to 37°C and thus inactivation of Mgm1 led to a drastic loss of cristae (*Figure 2A and B*). We expected that a time period of 25 min, which is much less than one generation time of yeast, would be too short to result in loss of mtDNA. Indeed, staining of mtDNA and test on respiratory competence revealed no loss of functional mtDNA upon exposure to 37°C for 25 min (*Figure 2C* and *Figure 2—figure supplement 3*). However, longer exposure (72 hr) of *mgm1-5* cells to non-permissive temperature led to inhibition of cell growth on respiratory medium (*Figure 2—figure supplement 3*). Strikingly, upon return of the *mgm1-5* cells to 25°C for 30 min cristae reappeared and septa were reduced, comparable to the situation before incubation at non-permissive temperature (*Figure 2A and B*). Interestingly, mitochondrial respiratory complexes in both WT and mutant, as determined for Complex III and IV, remained intact during the temperature shifts (*Figure 2—figure supplement 4*).

In parallel to the ultrastructure of the IM, the morphology of mitochondria was determined by fluorescence microscopy. While the normal mitochondrial tubular network was not altered by the temperature shift in WT, it was converted to numerous small fragments in almost all *mgm1-5* cells that were exposed to 37°C. Normal mitochondrial morphology was largely restored in *mgm1-5* cells during recovery at 25°C (*Figure 2D*).

Although the structural analysis of the *mgm1-5* mutant strongly suggested that crista formation and maintenance depends on Mgm1, reproducibly about 20% of the mitochondrial cristae remained intact after inactivation of Mgm1 in the various experiments (*Figure 2B*). To analyze the structure of these cristae we used electron tomography. Notably, exposure of *mgm1-5* cells to 37°C led to a

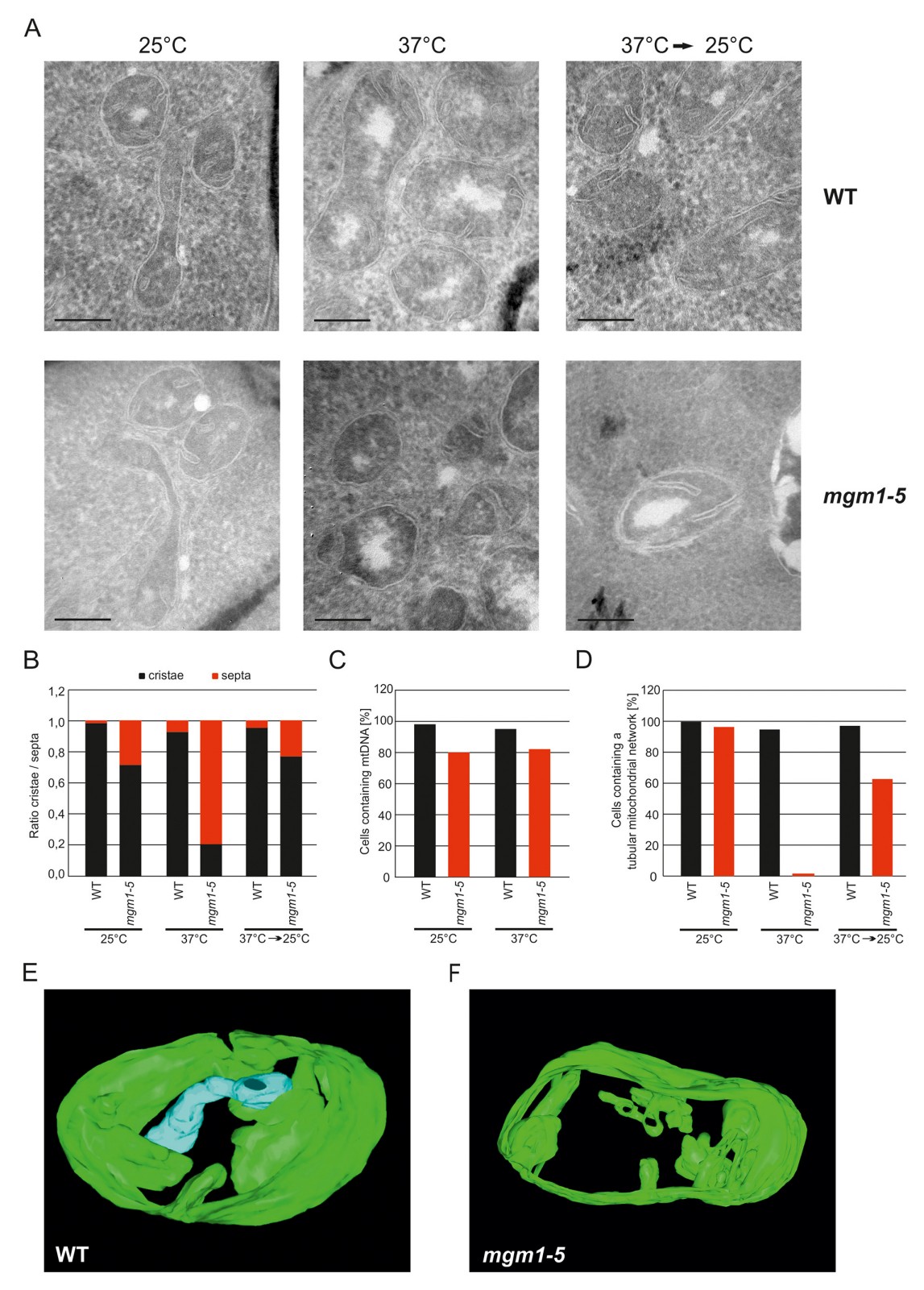

**Figure 2.** Mgm1 controls mitochondrial ultrastructure. (**A**) Inactivation of Mgm1 leads to rapid loss, and reactivation to the rapid regeneration of cristae. WT cells and cells expressing the temperature sensitive *mgm1-5* mutant were grown in YPD medium at 25°C to logarithmic phase. Aliquots of the cultures were incubated for 25 min at either 25°C or 37°C; further aliquots were incubated for 25 min at 37°C and shifted back to 25°C for 30 min. Cells were analyzed by EM. Scale bars, 0.2 μm. (**B**) Quantitative evaluation. 150–200 mitochondrial profiles were analyzed for each sample. (**C**) Maintenance of

*Figure 2 continued on next page*

*Figure 2 continued*

mtDNA in the *mgm1-5* mutant upon exposure to 37°C. WT and *mgm1-5* cells were grown in YPD medium at 25°C and incubated at 37°C for 25 min. The percentage of cells containing mtDNA was determined by DAPI staining. (D), Mitochondrial morphology in WT and in the *mgm1-5* cells expressing mitochondrially targeted GFP. Cells were treated as described in (A). The morphology of the mitochondrial network in 100 cells per sample was analyzed by fluorescence microscopy. (E) EM tomographic reconstruction of a mitochondrion of a WT yeast cell. Green, IBM and lamellar cristae connected to the IBM; blue, tubular crista. (F) Tomographic reconstruction of a mitochondrion of a *mgm1-5* cell grown at 25°C and shifted to 37°C for 25 min. Green, IBM and cristae connected to the IBM.

The following figure supplements are available for figure 2:

**Figure supplement 1.** Mgm1 is required for wild type inner membrane structure.

**Figure supplement 2.** Mgm1 is required for wild type protein composition.

**Figure supplement 3.** Incubation of *mgm1-5 cells* for 25 min at non-permissive temperature does not lead to loss of functional mtDNA.

**Figure supplement 4.** Changes of mitochondrial ultrastructure upon inactivation of Mgm1 do not affect the assembly state of respiratory chain supercomplexes.

decrease of lamellar and an increase of tubular cristae as compared to WT (*Figure 2E and F*; *Videos 1* and *2*). These results show that the disappearance and reappearance of cristae are rapid and depend on functional Mgm1. We conclude that the formation of lamellar and tubular cristae relies on two different pathways. Moreover, Mgm1 plays a direct role in the formation and maintenance of lamellar but not of tubular cristae.

To further study the role of Mgm1 in cristae formation we first explored where Mgm1 is exactly located in mitochondria. Quantitative immuno-EM revealed that the majority of Mgm1 was localized to the OM/IBM and only about one third to the matrix/crista space (*Figure 3A*). Mgm1 is present in two isoforms (*Shepard and Yaffe, 1999*; *Wong et al., 2000*). l-Mgm1 contains a transmembrane helix that anchors it to the IM; in contrast, s-Mgm1 lacks the transmembrane helix and was reported to be present at the IBM or OM (*Sesaki et al., 2003*; *Wong et al., 2000*) and a short form of Opa1 was found to be co-localized with the OM fission machinery (*Anand et al., 2014*). Importantly, both l-Mgm1 and s-Mgm1 forms are required for Mgm1 function in mitochondrial fusion (*DeVay et al., 2009*; *Zick et al., 2009*). To test where s-Mgm1 is located, we generated sub-mitochondrial vesicles by sonication and separated them by gradient density centrifugation. Remarkably, s-Mgm1 co-fractionated with the outer membrane marker Tom70, indicating that it is associated with the outer membrane (*Figure 3B*). These data suggest that Mgm1 could interact with the OM in the course of IM fusion. Thus, as outlined below, it is possible that after fusion of the OM a septum is formed by IMs that are not yet fused, and that Mgm1 at IM/OM contact sites initiates the conversion of these

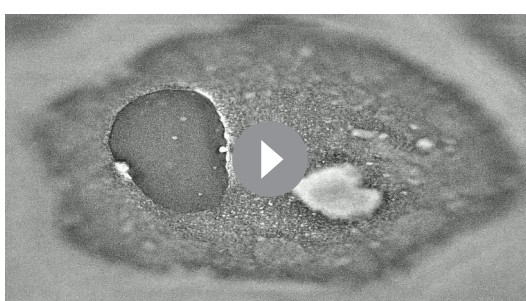

**Video 1.** 3D reconstruction with modelling of a mitochondrion in wild type cells. Grey, OM; green, IBM and cristae connected to the IBM; blue, tubular cristae without visible connection to the IBM.

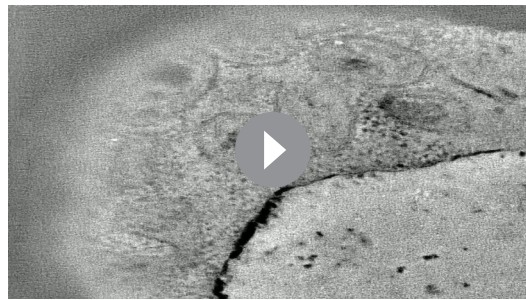

**Video 2.** 3D reconstruction with modelling of a mitochondrion in *mgm1-5* cells incubated for 25 min at 37°C. Grey, OM; green, IBM and cristae connected to the IBM.

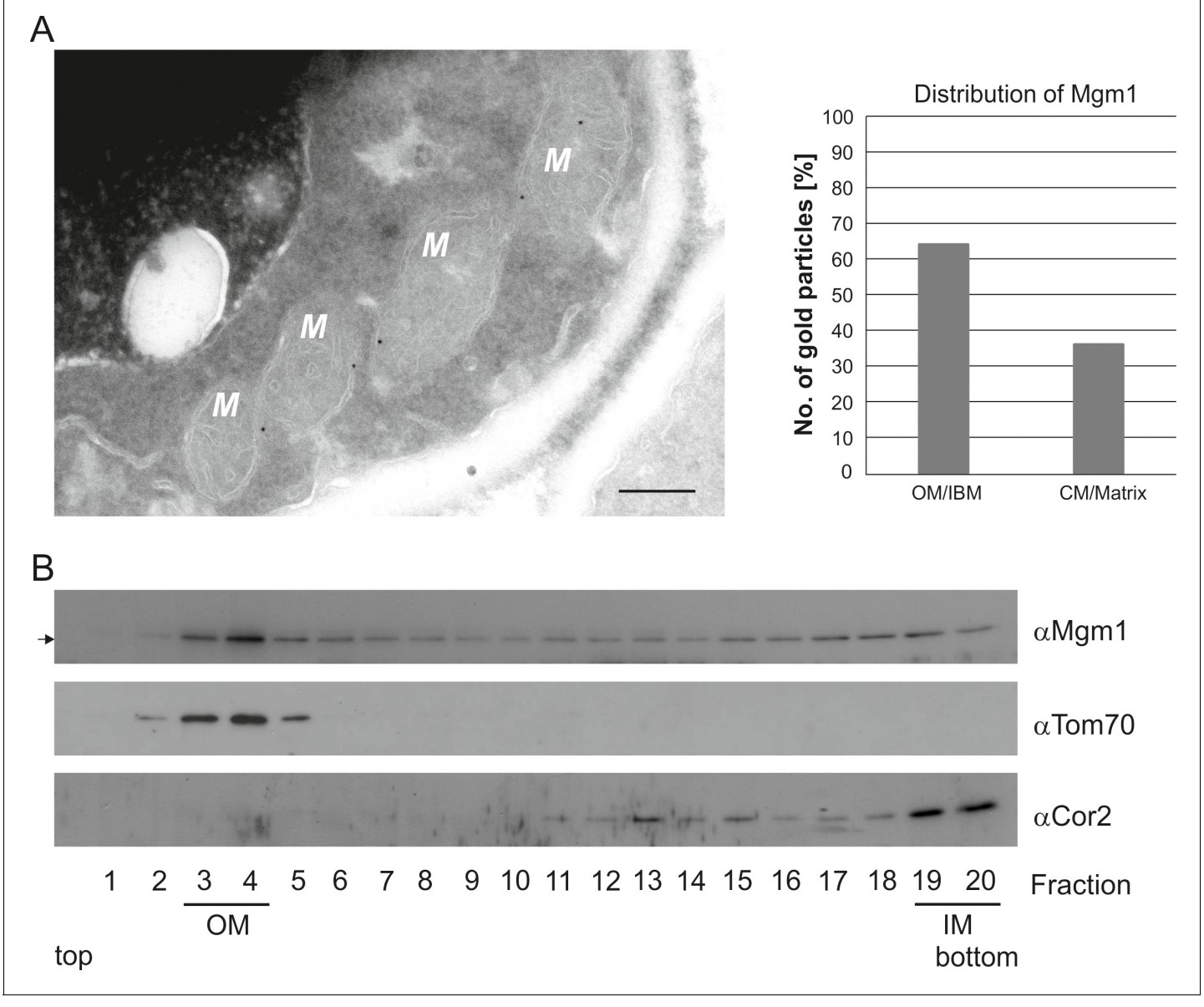

**Figure 3.** Mgm1 is present at the mitochondrial inner and outer membranes. (**A**) Distribution of Mgm1 between OM/IBM and cristae. Cells expressing 3xHA tagged Mgm1 were analyzed by immuno-EM. Left panel, four mitochondrial profiles (M) showing gold particles at the IBM. Scale bar, 0.2 μm. Right panel, quantification of gold particles. (**B**) Distribution of Mgm1 between OM and IM. Mitochondrial vesicles generated by sonication were subjected to sucrose density gradient centrifugation followed by fractionation, SDS-PAGE and immunoblotting. Arrow, s-Mgm1. The membrane anchor of l-Mgm1 is notorious for being clipped off during isolation of mitochondria (c.f. *Figures 5C* and *7C*). Therefore, l-Mgm1 was not detected after mitochondrial subfractionation. Cor2, subunit two of respiratory chain complex III.

septa membranes into a crista membrane by IM fusion.

## Dimeric F₁Fₒ-ATP synthase is preferentially located at crista membranes, but the monomeric form is present at septa and inner boundary membrane

Dimeric F$_1$F$_O$ provides positive curvature to crista membranes (*Dudkina et al., 2005*; *Strauss et al., 2008*; *Thomas et al., 2008*). The subunits of the F$_O$ sector, Su e/Atp21 and Su g/Atp20, mediate the formation of F$_1$F$_O$ dimers (*Arnold et al., 1998*; *Habersetzer et al., 2013*). Deletion of Su e or Su g leads to the formation of membrane sheets that are crossing the matrix space completely and to

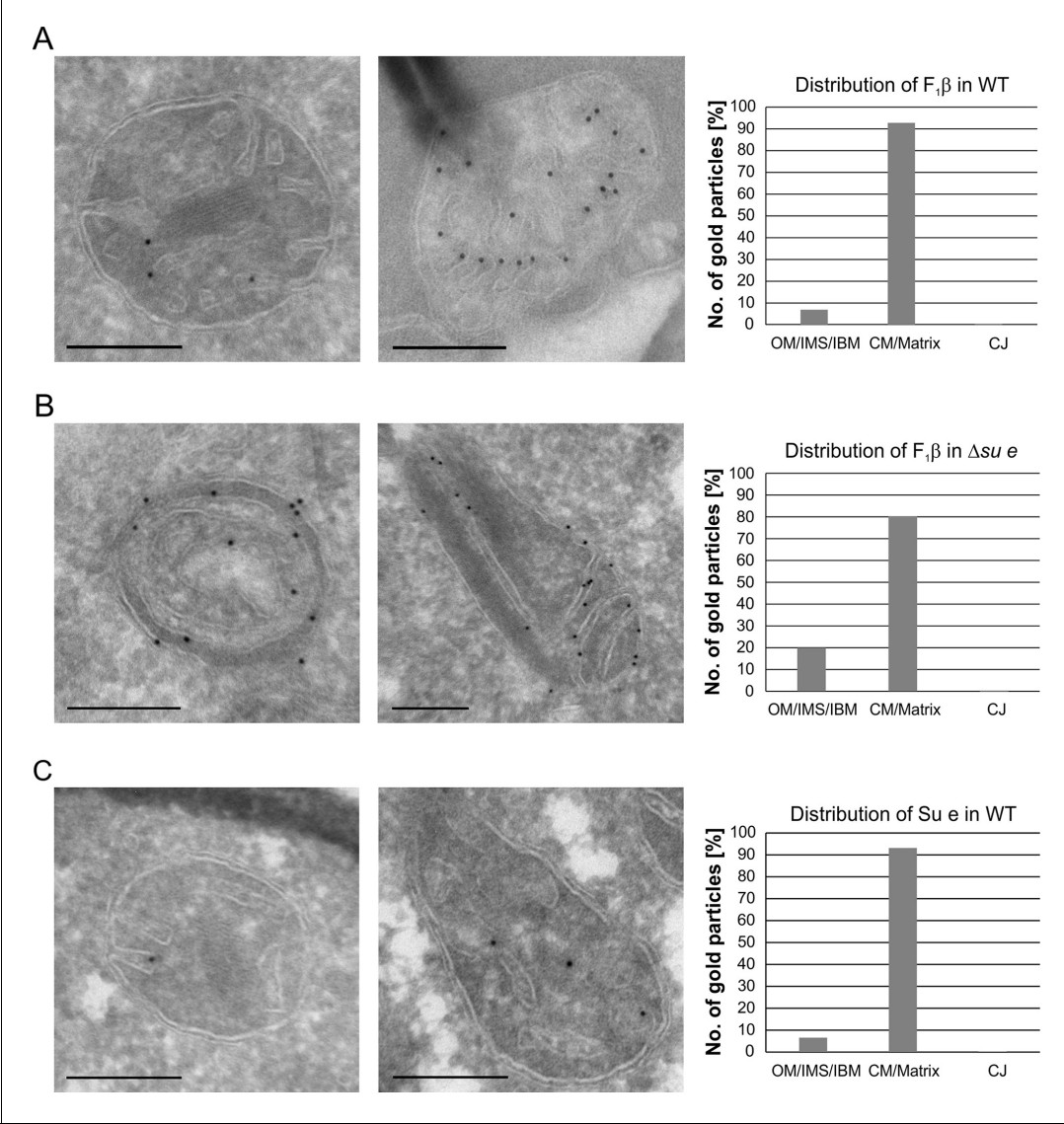

**Figure 4.** Localization and dimerization of $F_1F_O$ are relevant to mitochondrial architecture. (**A**) Distribution of $F_1F_O$ between IBM and cristae in WT cells. WT cells expressing 3xHA tagged $F_1\beta$ were subjected to immuno-EM and quantified. (**B**) Distribution of $F_1F_O$ between IBM and cristae/septa in $\Delta su\ e$ cells. $\Delta su\ e$ cells expressing 3xHA tagged $F_1\beta$ were analyzed as in (**A**). (**C**) Distribution of Su e between IBM and cristae in WT. WT cells expressing 3xHA tagged Su e were analyzed as in (**A**). Scale bars, 0.2 µm.

onion-like structures. Examination of a large number of EM sections showed that these membrane sheets are not cristae as previously assumed, but septa. Onions represent cross sections of curved septa (*Figure 1*). This is also apparent upon tomographic reconstruction of isolated mitochondria from cells lacking dimeric $F_1F_O$ (*Davies et al., 2012*). A fundamental difference between cristae and septa is that crista membranes are strongly bent to generate crista rims and thus the sac-like crista structure, whereas septa are not closed but divide the matrix into membrane-limited subcompartments. In spite of this drastic alteration of mitochondrial architecture, growth is only moderately retarded under respiratory conditions (*Paumard et al., 2002*; *Rabl et al., 2009*). It is presently unknown how and when during crista formation this bending is taking place. We localized fully assembled $F_1F_O$ by detection of its subunit $F_1\beta$ and found it almost exclusively in the crista membranes of WT mitochondria (*Figure 4A*). In the Su e deletion mutant, however, the fraction of monomeric $F_1F_O$ detected in the IBM was increased by a factor of three in comparison to WT (*Figure 4B*).

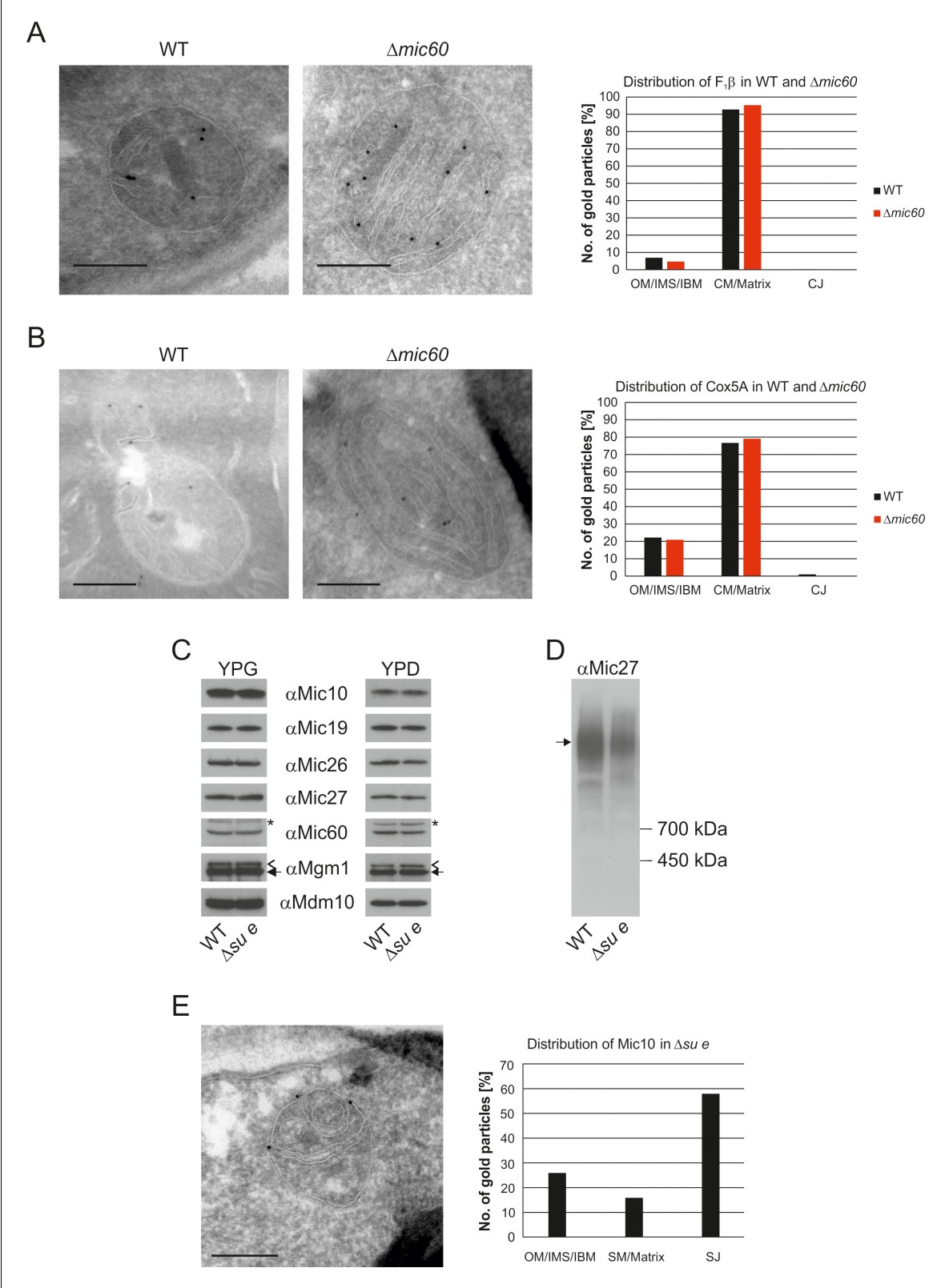

**Figure 5.** The stacked membrane sheets in mitochondria lacking Mic60 are functionally similar to crista membranes. (**A**) Distribution of $F_1\beta$ in $\Delta mic60$ cells. $\Delta mic60$ cells expressing 3xHA tagged $F_1\beta$ were analyzed by immuno-EM and quantified. (**B**) Distribution of Cox5A in $\Delta mic60$ cells. $\Delta mic60$ cells expressing 3xHA tagged Cox5A were analyzed as in (**A**). (**C**) Levels of MICOS subunits in $\Delta su\ e$ cells. WT cells and cells lacking Su e were grown on YPG (left) or YPD (right). Aliquots of mitochondrial protein were analyzed by SDS-PAGE and immunoblotting. Arrow head, l-Mgm1; arrow, s-Mgm1; asterisk,

*Figure 5 continued*

cross-reaction of the Mic60 antibody. (D) Assembly state of MICOS subunits in Δ*su e* cells. Mitochondria were isolated and lysed using digitonin. MICOS complex was analyzed by BN-PAGE and immunoblotting. Arrow, assembled MICOS complex. The result of one of four independently performed experiments is shown. (E) Distribution of Mic10 in Δ*su e* cells. Δ*su e* cells expressing 3xHA tagged Mic10 were analyzed as in (A). SM, septum membrane; SJ, septum junction. Scale bars, 0.2 μm.

This observation is compatible with the assumption that the distribution of $F_1F_O$ in cristae and IBM correlates with its oligomeric state. Similar to $F_1\beta$, Su e was present in crista membranes of WT mitochondria (*Figure 4C*). Very low levels of $F_1\beta$, monomeric $F_1F_O$ and unassembled Su e may exist in the IBM into which new components are continually being inserted, but dimeric $F_1F_O$ is present almost exclusively in cristae and does not assemble in the planar IBM. Thus, the assembly of dimeric $F_1F_O$ is likely to take place during the formation of cristae when the bending of membranes has to occur.

## The internal membranes in MICOS deficient mitochondria are structurally and functionally similar to crista membranes

The MICOS complex is located in the mitochondrial IM of yeast and higher eukaryotes, where it is required for the formation of CJs as well as the formation of contact sites with the OM. Mitochondria lacking Mic60 contain characteristic stacked membranes in the matrix that have no connections to the IBM (*Alkhaja et al., 2012*; *Darshi et al., 2011*; *Harner et al., 2011*; *Hoppins et al., 2011*; *Jans et al., 2013*; *John et al., 2005*; *Rabl et al., 2009*; *von der Malsburg et al., 2011*). Since CJs are essential elements of cristae, we asked whether MICOS has a function in the formation of cristae. A so far unanswered question is whether the cristae in WT and the stacked membranes in the Δ*mic60* mutant differ in their functions. The protein composition of these mitochondria is very similar to that of WT cells (*Harner et al., 2014*). Importantly $F_1F_O$ is present in the dimeric form (*Rabl et al., 2009*). To analyze whether there is a difference in the distribution of IM proteins in the stacked membranes in the Δ*mic60* mutant as compared to WT cristae, we examined the distribution of $F_1\beta$ and the cytochrome c oxidase subunit Cox5a by immuno-EM. $F_1\beta$ was found almost exclusively in the IM stacks of Δ*mic60* mitochondria as in WT (*Figure 5A*). Likewise, the distribution of Cox5a between the IBM and the cristae in WT cells and between IBM and the internal membranes in the Δ*mic60* mutant was almost identical (*Figure 5B*). The fact that the distribution of these two proteins is not changed in mitochondria of Δ*mic60* cells suggests that CJs are not required for assembly of protein complexes in the mitochondria and integration into crista membranes. Consistently, mitochondria of the Δ*mic60* mutant are able to perform oxidative phosphorylation (OXPHOS) and thus maintain mtDNA.

Deletion of Su e did not lead to a reduction of MICOS subunits both during respiratory growth and fermentative growth (*Figure 5C*). The level of the assembled MICOS complex, however, was strongly diminished (*Figure 5D*). This suggests that MICOS is partly dissociated when cristae are absent. Interestingly, the MICOS subunit Mic10 was found to be strongly enriched at the sites where septa membranes meet the IBM and the bending of septa is strongest (*Figure 5E*). This observation is in line with the finding that this MICOS subunit has an important role in the formation of the narrow ring or slot like structure of the crista junctions (*Barbot et al., 2015*; *Bohnert et al., 2015*; *Milenkovic and Larsson, 2015*).

Taken together, this suggests that cristae and stacks are functionally similar if not identical. Furthermore, in the absence of the dimeric ATP synthase assembled MICOS is not sufficient to allow the generation of cristae. These findings raise the questions of how the internal membrane stacks originate and how proteins reach them in view of the virtually complete absence of connections with the IBM.

## Mitochondria lacking Dnm1 show altered crista structure

Dnm1 (in yeast)/Drp1 (in higher eukaryotes), a dynamin-related GTPase, is an essential component of the fission machinery. Its deletion leads to the formation of a highly branched and interconnected mitochondrial network (*Bleazard et al., 1999*; *Otsuga et al., 1998*; *Smirnova et al., 1998*). Loss of Mgm1 in cells depleted of Dnm1 does not lead to loss of cristae (*Sesaki et al., 2003*). This surprising

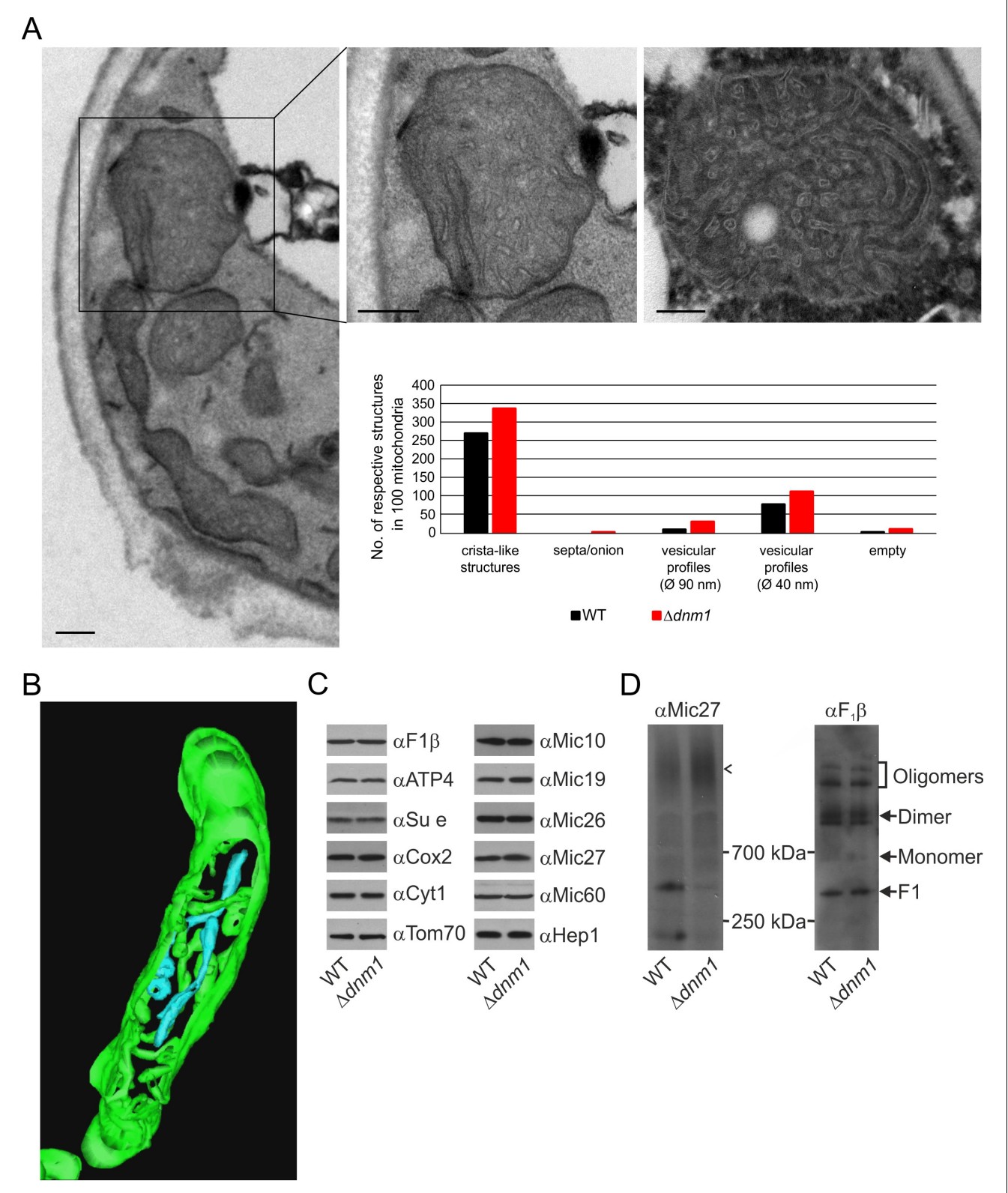

**Figure 6.** Structure and protein composition of cristae in cells deficient in Dnm1. (**A**) Ultrastructure and quantitative evaluation of mitochondria in Δ*dnm1* cells grown on YPG (scale bars, 0.2 μm). (**B**) Tomographic reconstruction of a mitochondrion of a Δ*dnm1* cell. Green, IBM and cristae connected to the IBM; blue, tubular cristae without visible connection to the IBM. (**C**) Steady state levels of mitochondrial proteins of Δ*dnm1* cells. WT and Δ*dnm1*

*Figure 6 continued*

cells were grown on YPG, aliquots of mitochondrial protein were analyzed by SDS-PAGE and immunoblotting. (D) Assembly state of MICOS and of $F_1F_O$ in Δ*dnm1* cells. Analysis as in **Figure 4D**. Arrow head, assembled MICOS complex.

observation appeared to contradict our suggestion that Mgm1 is directly required for crista formation and maintenance. To resolve this apparent discrepancy, we first analyzed the ultrastructure of mitochondria in the Δ*dnm1* mutant. We observed that mitochondrial architecture differed significantly from that in WT. These mitochondria are considerably enlarged, densely filled with vesicular profiles and short tubular cristae, while CJs are abundant along the IBM (**Figure 6A**). In comparison to WT, the number of cristae with lamellar structure was found to be much lower as highlighted by electron tomography. 3D reconstruction revealed that the vesicular profiles mostly represent interconnected highly convoluted and branched tubular structures (**Figure 6B**; **Video 3**). These tubules seem to have the ability to fuse and to branch. Interestingly, we identified tubular cristae also in WT cells by tomography, albeit much less frequently (**Figure 2E**; **Video 1**). Normal levels of components of OXPHOS as well as assembled MICOS and dimeric $F_1F_O$ were present in these mitochondria (**Figure 6C and D**). Notably, the level of the assembled MICOS complex in the Δ*dnm1* mutant was significantly increased. This is consistent with the higher number of cristae and CJs in this mutant. The necessity for the accommodation of OXPHOS and other membrane proteins likely requires an increase of the number of cristae since the surface area of tubular cristae compared to lamellar cristae is much lower. In summary, we reasoned that the mitochondrial network is in an almost completely fused state in the absence of Dnm1, and thus the rate of fusion is strongly decreased. The diminished Mgm1 activity might then result in a drastically reduced formation of lamellar cristae and massive production of tubular cristae.

## Mitochondrial fusion/fission dynamics contributes to the formation of lamellar cristae

In order to obtain more insight into the role of Mgm1 in the formation of cristae, we analyzed the biochemical composition and ultrastructure of mitochondria in the Δ*dnm1*Δ*mgm1* double deletion mutant. Growth of the Δ*dnm1* and Δ*dnm1*Δ*mgm1* strains on non-fermentable carbon sources was similar (**Figure 7A**). On fermentable carbon source, however, the double deletion mutant displayed a high rate of loss of mtDNA, pointing to a residual function of Mgm1 in mitochondria of the Δ*dnm1* mutant (**Figure 7B**). The steady state levels of a number of mitochondrial proteins were similar in WT, Δ*dnm1*, and Δ*dnm1*Δ*mgm1* strains (**Figure 7C**). The interior of Δ*dnm1*Δ*mgm1* mitochondria, like that of the Δ*dnm1* mutant, was full of tubular cristae. Strikingly, lamellar cristae were not observed in mitochondria of the Δ*dnm1*Δ*mgm1* mutant (**Figure 7D and E**; **Video 4**). Thus, Mgm1 appears to be essential for the formation of

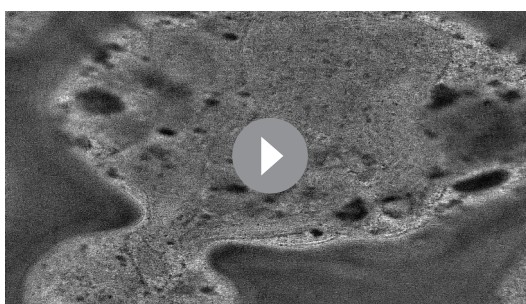

**Video 3.** 3D reconstruction with modelling of a mitochondrion in the Δ*dnm1* mutant. Green, IBM and cristae connected to the IBM; blue, tubular cristae without visible connection to the IBM; yellow, tubular cristae without connection to the IBM.

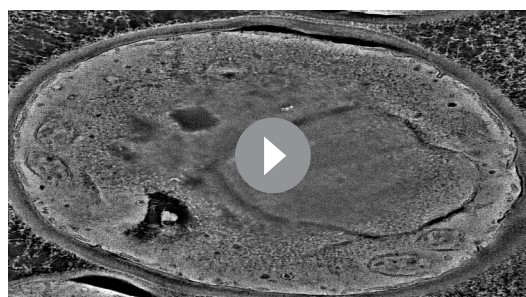

**Video 4.** 3D reconstruction with modelling of a mitochondrion in the Δ*dnm1*Δ*mgm1* double deletion mutant. Green, IBM and cristae connected to the IBM; blue, tubular cristae without visible connection to the IBM; yellow, tubular cristae without connection to the IBM.

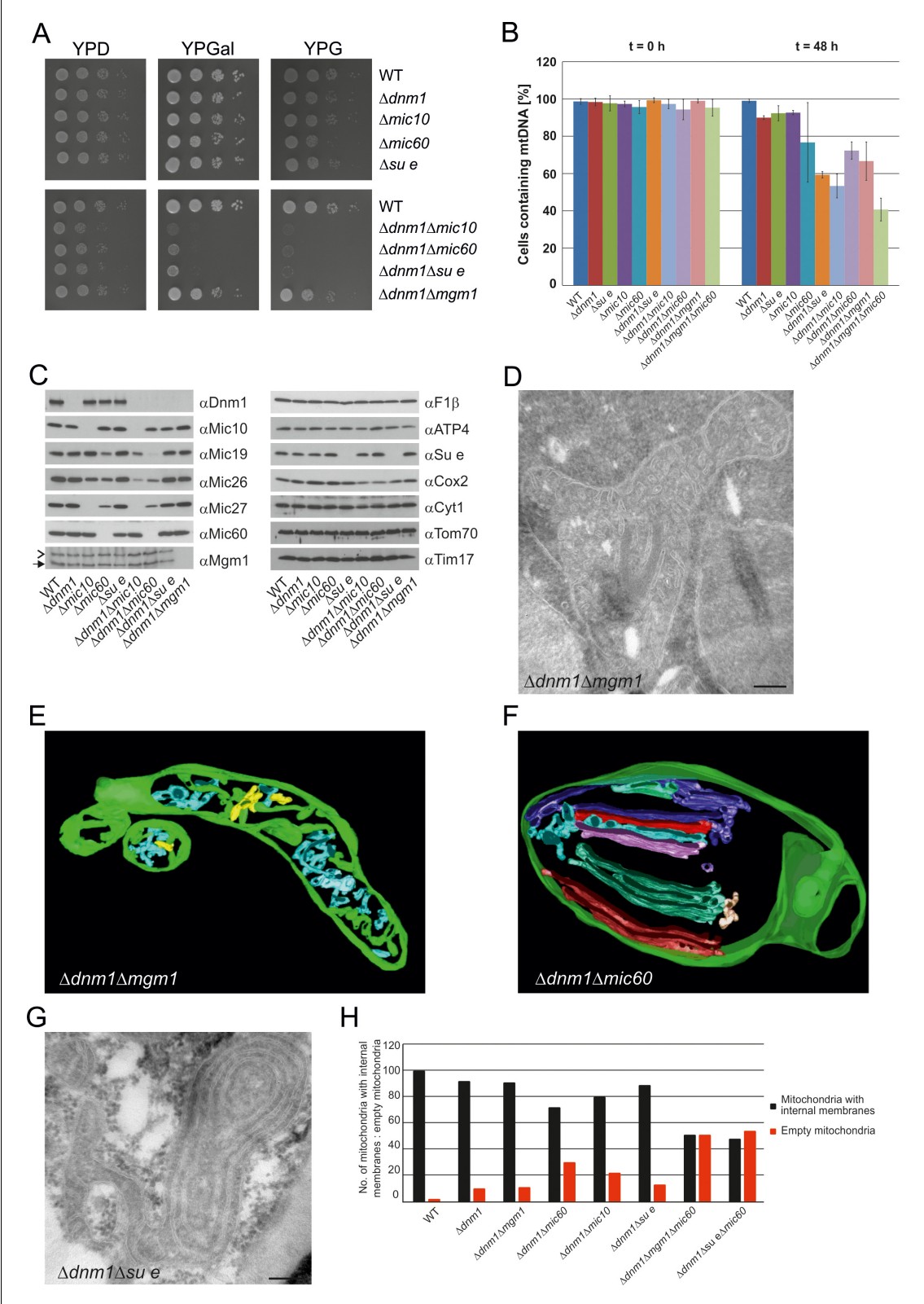

**Figure 7.** Growth, structure and protein composition of cristae in mutants lacking either Mgm1, Mic10, Mic60 or Su e in the Dnm1 deletion background. (**A**) Growth of double mutant strains deficient in Dnm1 and Mgm1, Mic10, Mic60 or Su e. The strains were cultured on YPG medium to logarithmic phase followed by growth analysis on the indicated media by drop dilution assay. (**B**) mtDNA maintenance in cells lacking either Mgm1, dimeric $F_1F_O$ or MICOS in the Δ*dnm1* background. Strains were grown on YPG and transferred to YPD medium. At time 0 hr and after 48 hr the
*Figure 7 continued on next page*

Figure 7 continued

percentage of cells containing mtDNA was determined by DAPI staining. Mean values of three independent experiments. Error bars, standard deviation. (C), Steady state levels of mitochondrial proteins in the Δdnm1 double mutant cells. Cells were grown on YPG medium, proteins were extracted and their levels were analyzed by SDS-PAGE and immunoblotting. Arrow head, l-Mgm1; arrow, s-Mgm1. (D) Mitochondrial ultrastructure of Δdnm1Δmgm1 cells analyzed by EM. (E) Tomographic reconstruction of a mitochondrion of a Δdnm1Δmgm1 cell. Green, IBM and cristae connected to the IBM; blue, tubular cristae without visible connection to the IBM; yellow, tubular cristae without connection to the IBM. (F) Tomographic reconstruction of a mitochondrion of a Δdnm1Δmic60 cell. Green, IBM; other colors, perforated tubular-sheet like membrane structures in the matrix without connections to the IBM. To be able to discriminate between the different tubular elements they are shown in different colors. (G) Ultrastructure of mitochondria in Δdnm1Δsu e cells. (H) Quantitative evaluation of the EM analysis of the indicated mutant strains. Scale bars, 0.2 μm.
The following figure supplement is available for figure 7:

**Figure supplement 1.** Mitochondrial outer membrane fusion is important but not essential for the formation of lamellar cristae.

lamellar cristae, but is not required for the formation of tubular cristae and CJs.

To further study the role of mitochondrial fusion in the formation of lamellar cristae, we analyzed mitochondrial architecture in △dnm1△fzo1 cells, which have a block of outer membrane fusion and fission. Indeed, we observed mitochondria with numerous tubular cristae (*Figure 7—figure supplement 1*), supporting the idea that tubular cristae are the predominant type in the absence of fusion activity. However, this phenotype was observed to a much lesser degree than in the △dnm1△mgm1 mutant. Strikingly, △dnm1△fzo1 cells quite frequently contained constricted mitochondria and mitochondria with septa (*Figure 7—figure supplement 1*). Since it has been repeatedly suggested that a separate Dnm1-independent fission mechanism exists for the IM (*Gorsich and Shaw, 2004*; *Ishihara et al., 2013*; *Westermann, 2010*), we consider it possible that these structures represent division events of the IM in the absence of OM fusion. This could then provide Mgm1 with suitable inner membrane structures for the formation of lamellar cristae and explain why the cristae phenotype of △dnm1△fzo1 is less severe than that of △dnm1△mgm1. Together, these observations suggest that mitochondrial fusion is important for generating crista structure.

We further investigated the mitochondria in the Δdnm1Δmic60 mutant; they displayed a virtually complete absence of cristae and CJs. Vesicular-tubular profiles, often located close to the IBM and associated with sheet-like membrane structures, were present in the interior of the mitochondria. Tomographic reconstructions revealed a rather corrugated and perforated appearance, in contrast to the planar structure of WT lamellar cristae. Therefore, these membranes are most likely formed by association of extended tubules, which partly fuse with each other (*Figure 7F*; *Video 5*). Interestingly, in contrast to the Δdnm1Δmgm1 mutant, growth of Δdnm1Δmic60 cells was very slow on respiratory medium (*Figure 7A*) and the rate of mtDNA loss was higher in this mutant, indicating that in the absence of fully functional Mgm1, MICOS and thereby CJs and tubular cristae contribute to maintenance of mtDNA (*Figure 7B*). The steady state levels of the mitochondrial proteins analyzed were similar to those in Δdnm1 cells. MICOS components were an exception, as they were reduced to different degrees. This was, however, also observed with the △mic10 or △mic60 single deletion mutants. The level of Cox2, a mitochondrial gene product, was also reduced (*Figure 7C*). We suggest that in Δdnm1Δmic60 mutant, in contrast to the Δmic60 single mutant in which lamellar cristae can be made, the proteins and lipids imported into the mitochondria invaginate the IBM. This results in irregular networks which, as indicated by the biochemical and the growth phenotype, lack the characteristic properties of cristae.

Deletion of both Dnm1 and Su e resulted in cells that grew extremely slowly on non-fermentable carbon source (*Figure 7A*). Mitochondria contained many highly branched septa and multi-layered onion-like profiles (*Figure 7G*). In addition, mtDNA was lost at a high rate, indicating that dimeric $F_1F_O$ is important for maintenance of mtDNA when Mgm1 has no obvious role (*Figure 7B*). Strikingly, mitochondria in the Δdnm1Δmgm1Δmic60 and the Δdnm1Δsu eΔmic60 triple mutants showed even more severe ultrastructural alterations than the Δdnm1Δmgm1, Δdnm1Δmic60, or Δdnm1Δsu e double mutants. Fewer internal sheet-like membranes were present and approximately 50% of mitochondrial profiles were completely empty (*Figure 7H*). Loss of mtDNA in the Δdnm1Δmgm1Δmic60 triple mutant was particularly high (*Figure 7B*). Attempts to generate a respiratory competent Δdnm1Δmgm1Δsu e triple or Δdnm1Δmgm1Δmic60Δsu e quadruple mutant by either strain crossing

or direct deletion were not successful, suggesting that the simultaneous loss of Mgm1, $F_1F_O$ dimers and MICOS is deleterious for crista formation and/or maintenance of mtDNA.

We conclude that the pathway of tubular crista formation does not need the fusion activity of Mgm1, but does require both MICOS and dimeric $F_1F_O$. In contrast, the pathway leading to the formation of lamellar cristae, in addition to MICOS and dimeric $F_1F_O$, appears to depend on Mgm1. Furthermore, the drastically altered membrane structures present in cells lacking Dnm1 and at the same time MICOS core components or Su e cannot substitute for lamellar and tubular cristae to maintain mtDNA and respiratory growth.

## Discussion

Major advances have been made in our understanding of the biogenesis of mitochondrial protein complexes and supercomplexes, but the biogenesis of mitochondrial architecture, the next higher level of organization, has been investigated to a much lesser degree. In particular, organization and formation of the cristae are poorly understood. We report here on novel and basic insights into the molecular mechanisms underlying the formation of lamellar and of tubular cristae, and the key factors involved in these pathways.

Three factors have decisive functions in the biogenesis of cristae: Mgm1, the dimeric $F_1F_O$-ATP synthase, and the MICOS complex. Based on our findings and the literature about these three factors and their interactors, we developed a hypothesis that postulates the existence of two distinct mechanisms for the generation of lamellar and tubular cristae. Regarding the formation of lamellar cristae, a first important issue is how Mgm1 performs fusion of the mitochondrial IM (*Figure 8A*). It is known that dysfunction of the components that mediate mitochondrial membrane fusion, Fzo1, Ugo1, and Mgm1, leads to loss of mtDNA and loss of cristae, concomitant with septa formation and inhibition of IM fusion (*Hermann et al., 1998*; *Hoppins et al., 2009*; *Meeusen et al., 2006*; *Sesaki et al., 2003*). It was shown that Fzo1 in the OM interacts with the IM, and Mgm1 in the inner membrane interacts physically with Fzo1 and Ugo1 in the OM (*Fritz et al., 2001*; *Sesaki et al., 2003*; *Wong et al., 2003*). Both l-Mgm1 anchored in the IM and s-Mgm1 bound to the OM are required for fusion (*DeVay et al., 2009*; *Zick et al., 2009*). These observations suggest a mechanism by which progression of IM fusion takes place at sites where IM and OM are in close contact (*Figure 8A*). Fusion of the OM by Fzo1 would first generate a mitochondrion with a planar septum consisting of non-fused IM. Tethering of these two IMs by Mgm1 could take place at the IM surface in contact with the OM. According to our hypothesis, the fusion of the IM is then initiated by Mgm1 and proceeds along the IM-OM contacts, thereby generating a membrane sac protruding into the matrix. Shifting the temperature sensitive *mgm1-5* mutant to non-permissive temperature leads to rapid fragmentation of mitochondria (*Meeusen et al., 2006*; *Wong et al., 2000*). Quantitative EM analysis of this process reveals that, in addition to fragmentation, a rapid and extensive loss of cristae takes place, while septa are formed. These findings strongly suggest that the activity of Mgm1 not only is required for fusion but also for maintenance of cristae. Importantly, cristae are rapidly regenerated upon reactivation of Mgm1. Equally important, mtDNA is retained during inactivation of Mgm1 and remains functional. Thus, it is not the loss of mtDNA that causes loss of cristae and formation of septa, emphasizing a role of Mgm1 in crista maintenance.

At the same time these findings raise the question how this rapid reformation of cristae can be explained. The fusion rate of mitochondria in wild type cells has been measured by live cell microscopy (*Jakobs et al., 2003*). Under steady-state conditions at least one fusion and one fission event was observed per minute and cell. This rate is apparently determined by regulatory processes that ensure a balance of fusion and fission. In contrast, fusion of the IM in *mgm1-5* mutant cells by reactivation of Mgm1 is likely to reflect an entirely different situation. Here, the conversion of a septum to a crista requires only the time of IM fusion by Mgm1 which upon its reactivation is immediately ready to function. The overall process of mitochondrial fusion requires a multitude of steps, including fusion of the OM. Fusion of the IM likely represents one of the fastest steps. Notably in this context, fusion intermediates have not been identified convincingly in intact cells. These conclusions are in good agreement with observations that acute ablation of Opa1 in mouse embryonic fibroblasts leads to disorganized cristae and reduced respiratory function, but maintenance of mtDNA (*Cogliati et al., 2013*).

The conversion of two membranes into a single one by Mgm1 entails the necessity to bend the membrane. Very interestingly, recent experiments with reconstituted vesicles have revealed that

Mgm1 mediates membrane bending (*Rujiviphat et al., 2015*). Bending of crista membranes takes place only when the $F_1F_O$-ATP synthase can dimerize (*Davies et al., 2012*; *Rabl et al., 2009*). Therefore, it is reasonable to assume that bending of crista membranes takes place concurrently with the fusion process, and dimeric $F_1F_O$-ATP synthase is required to stabilize the bending of the membrane. Importantly, according to our model, in the absence of Su e, a dimerization component of the $F_1F_O$-ATP synthase, fusion of IM by Mgm1 cannot take place. Rather, OM fusion still occurs in the absence of IM fusion and septa are generated. The continual import of proteins and lipids then leads to expansion of the septa membranes. This expansion may cause the curved structures of septa that in EM sections appear as onion-like profiles (*Figure 1*).

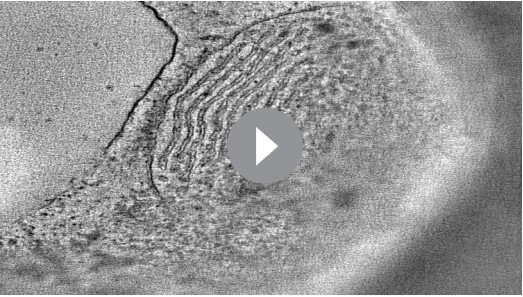

**Video 5.** 3D reconstruction with modelling of a mitochondrion in the Δ*dnm1*Δ*mic60* double deletion mutant. Green, IBM and cristae connected to the IBM; blue and red, fused tubular structures and perforated tubular-sheet like membrane structures in the matrix without connections to the IBM; yellow, ER-type tubules.

Our hypothesis proposes that fusion of IM is halted by the assembly of MICOS complexes, leading to the generation of a CJ and thereby represents the final step in the formation of lamellar cristae (*Figure 8A*). In agreement with this speculation, we found that the MICOS component Mic10 is preferentially located at sites where septa merge with the IMS, the septum junctions. Our hypothesis does not exclude a stabilizing function of MICOS for cristae. Since membrane proteins and protein complexes can shuttle between cristae and the IBM (*Vogel et al., 2006*), it seems likely that a major reason for crista stability is the continued presence and activity of Mgm1. In addition, our hypothesis provides a rational explanation for the generation of the stacked closed membrane sheets lacking connection with the IBM that are formed in the absence of functional MICOS complex. In the absence of the MICOS core components, Mic60 and Mic10, fusion proceeds as in WT cells, but cannot be halted (*Figure 8A*). Interestingly, a very similar mechanism was suggested for the fusion of vacuoles, the yeast homolog of lysosomes (*Wickner, 2010*). Upon fusion of vacuoles an internal vesicle is generated (*Wang et al., 2002*). It might well be that these vesicles are generated like the internal stacks in Mic10 or Mic60 deletion mutants. According to our hypothesis, during fusion septa membranes and the nascent crista membrane form a continuum. This allows free distribution of proteins between the IBM and the nascent crista membrane. This would also explain the similarity of the composition of intramitochondrial membrane sheets, made in the absence of MICOS, to crista membranes as well as their full functionality in mediating OXPHOS.

After completion of fusion and formation of CJs, cristae may grow by uptake of newly synthesized proteins, along with assembly of subunits of the $F_1F_O$-ATP synthase, which extend the crista rims. Thus, our hypothesis also offers an explanation of how the characteristic shape of lamellar cristae is generated and how cristae can expand.

Our results also suggest the existence of a second pathway of crista biogenesis (*Figure 8B*). In this pathway, import of newly synthesized proteins and lipids into the IBM and passage through CJs leads to the formation of tubular cristae. Assembly of CJs can take place since a pool of MICOS subunits is present in the IBM (*Harner et al., 2011*). Interestingly, oligomeric forms of the MICOS subunit Mic10 can induce membrane curvature in vitro and were suggested to form CJs in vivo (*Barbot et al., 2015*; *Bohnert et al., 2015*; *Milenkovic and Larsson, 2015*). We propose that the shape of CJs formed by the MICOS complex determines the tubular shape of nascent cristae. Dimerization of $F_1F_O$ would stabilize bending of the tubular cristae, very much like it stabilizes the rims of lamellar cristae. Interestingly, regular helical zipper-like structures of presumably dimeric $F_1F_O$ were observed at the surface of the tubular cristae of the protist *Paramecium multimicronucleatum* by scanning EM (*Allen et al., 1989*). Remarkably, in the absence of both Dnm1 and functional MICOS, tubules are formed. We suggest that they are generated by the continual influx of proteins and lipids and shaped by dimeric $F_1F_O$. Since CJs are not formed, these tubules are rather unstable, having a high tendency to fuse and to generate bizarre corrugated networks, and are not functional in

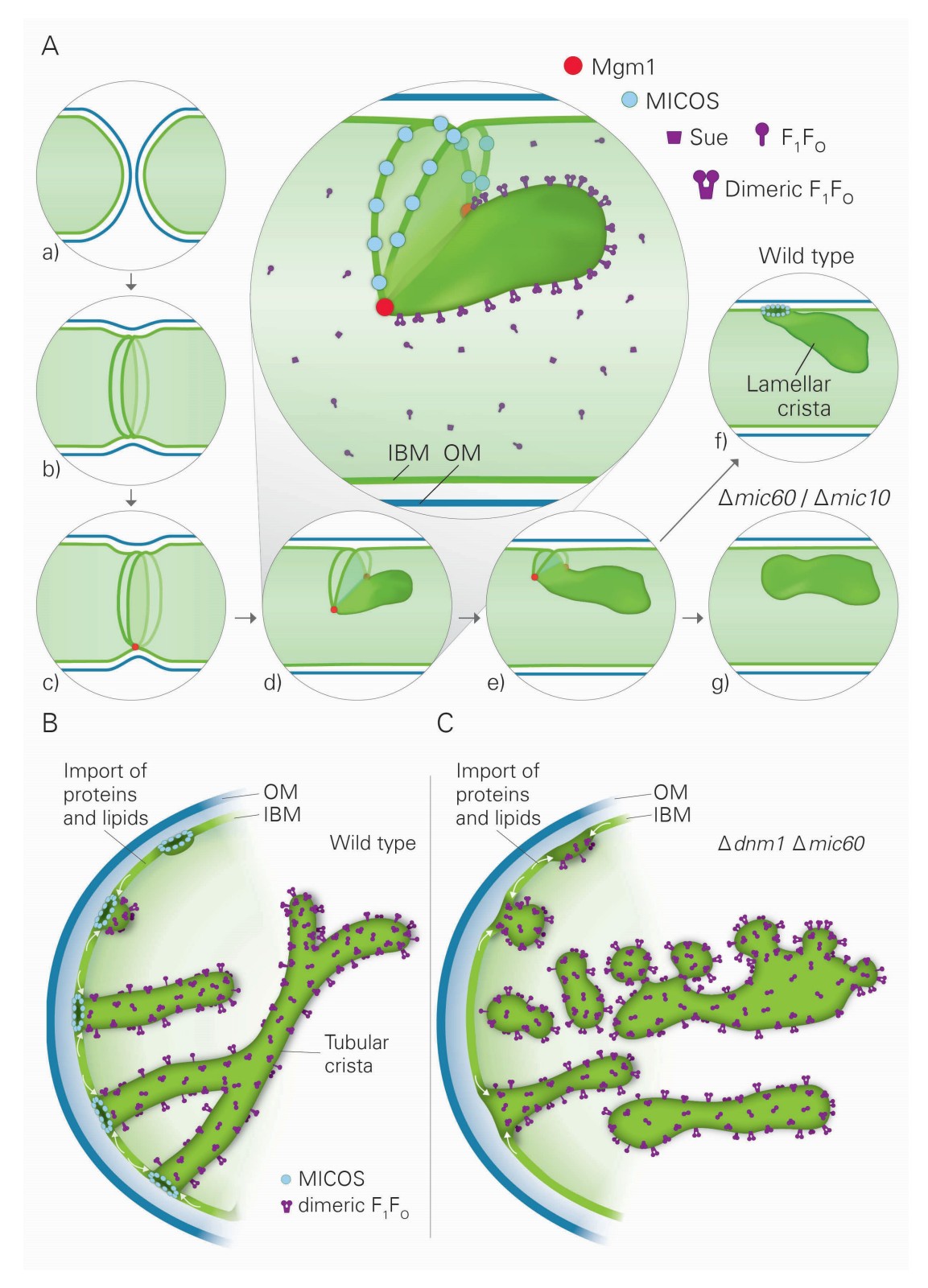

**Figure 8.** Model of the formation of cristae by the fusion dependent and independent pathways. (**A**) Formation of lamellar cristae by the fusion dependent pathway. Steps in the conversion of the septa membranes of two fusing mitochondria into a crista membrane. (**a**) Two mitochondria before fusion; (**b**) after fusion of OM and before fusion of IM; (**c**) tethering of the two septa membranes; (**d–e**) progressing fusion intermediates; (**f**) final stage of fusion after formation of a crista junction; (**g**) formation of a closed crista-like membrane vesicle without connection to the IBM in mutants deficient in

*Figure 8 continued on next page*

*Figure 8 continued*

Mic60 or Mic10. (**B**) Formation of tubular cristae by the fusion independent pathway. (**C**) Formation of bizarre membrane assemblies in mitochondria of MICOS deficient cells by the fusion independent pathway. Blue lines, OM; green lines, IBM, septa membrane and crista membrane; light green, matrix space.

OXPHOS and maintenance of mtDNA (*Figure 8C*). This points to a correlation between presence of cristae and mtDNA. In fact, mtDNA, condensed in nucleoids, was reported to be bound to crista membranes, possibly to crista rims (*Kopek et al., 2012*; *Kukat and Larsson, 2013*).

This second pathway of crista biogenesis suggested in our hypothesis becomes apparent when fission is compromised. In this situation, the mitochondrial network is virtually completely fused and Mgm1-dependent fusion activity should be drastically reduced. The lack of requirement of Mgm1 in this pathway provides a rational explanation for the most surprising observation that Mgm1 can be deleted in the Dnm1 deletion background without loss of cristae. Residual formation of lamellar cristae in the absence of Dnm1 can be explained by residual fusion and fission activity of mitochondria (*Gorsich and Shaw, 2004*; *Ishihara et al., 2013*; *Westermann, 2010*). The virtually complete lack of lamellar cristae in cells deficient in both Dnm1 and Mgm1 further supports this assumption. Most interestingly, a study on the role of Opa1 in mitochondrial cristae remodeling showed that mitochondria isolated from mouse embryonic fibroblasts lacking functional Opa1 lacked lamellar cristae; instead, these mitochondria contained short tubular-type cristae (*Frezza et al., 2006*). These observations are not only in very good agreement with our conclusions on a direct role of Mgm1 in crista formation, but also point to the conservation of its role in crista biogenesis in mammalian cells.

Can both pathways of crista biogenesis coexist? This is clearly the case in yeast, as we observed lamellar and, to a lower extent, tubular cristae in WT cells. Inspection of a large number of published EM images of mitochondria from different cell types reveals a bias to lamellar vs. tubular cristae. For example, lamellar cristae dominate in pancreatic cells and hepatocytes, whereas in steroidogenic cells tubular cristae are the principle architectural feature. A mixed situation prevails in certain types of muscle cells where both lamellar and tubular profiles can be recognized (*Blinzinger et al., 1965*; *Fawcett, 1981*; *Hanaki et al., 1985*). Tomographic reconstruction is the optimal approach to determine precisely the distribution of these two types of cristae. Such a study has been performed with cerebellar neuronal cells and revealed lamellar and tubular cristae that are regularly connected by sites of fusion (*Perkins et al., 1997*). Furthermore, the differentiation of isolated zona glomerulosa cells of rat adrenal cortex triggered by treatment with adrenocorticotropic hormone leads to a drastic change of the corticoid hormones synthesized and to a concomitant transformation of the crista structure from mainly "lamellar-tubular" to "tubular-convolute" (*Andreis et al., 1990*). Interestingly, steroidogenesis in human trophoblasts was reported to be dependent on the activity of Opa1. During developmental induction of the steroidogenic pathway, Opa1 levels decreased and steroid formation increased. Manipulations of cells to increase Opa1 levels led to decreased efficiency of steroidogenesis. At the same time, crista structure was observed to be remodeled (*Wasilewski et al., 2012*). Altogether, many basic questions need to be answered, including how mitochondrial architecture is modulated in response to metabolic and developmental requirements, how the state of cellular differentiation affects mitochondrial ultrastructure and why specific crista structures are necessary for specific cellular functions.

In conclusion, our investigation of the determinants of mitochondrial architecture has led us to an entirely novel concept of pathways of mitochondrial biogenesis. Since the identified key players are highly conserved from yeast to human, our study has immediate relevance for the relationship of structure and function of mammalian mitochondria of different tissues under normal and dysfunctional conditions. Finally, our results emphasize that understanding inheritance and homeostasis of mitochondria requires an integrated view on their three-dimensional architecture and the molecular topological functions of the various key players involved.

**Table 1.** Genotypes of the strains used in the study. Left column, names of the stains. Right column, genotypes of the strains.

| Strain | Genotype |
| --- | --- |
| W303 WT | *MATa or MATa, ade2-1, leu2-3, his3-11,15, trp1-1, ura3-1, can1-100* |
| W303 mgm1-5 | *MATa, ade2-1, leu2-3, his3-11,15, trp1-1, ura3-1, can1-100, mgm1-5 (G408 to D408)* (*Wong et al., 2000*) |
| YPH499 WT | *MATa; ura3-52, lys2-801amber, ade2-101ocre, trp1-Δ63, his3-Δ200, leu2-Δ1} or BY4724 {MATα; his3Δ1; leu2Δ0; lys2Δ0; ura3Δ0* |
| Δsu e | *MATa; ura3-52, lys2-801amber, ade2-101ocre, trp1-Δ63, his3-Δ200, leu2-Δ1, su e::HISI3* |
| F1β-3xHA | *MATa; ura3-52, lys2-801amber, ade2-101ocre, trp1-Δ63, his3-Δ200, leu2-Δ1, F1β-3xHA::HISI3* |
| Δsu e F1β-3xHA | *MATa; ura3-52, lys2-801amber, ade2-101ocre, trp1-Δ63, his3-Δ200, leu2-Δ1, su e::HISI3, F1β-3xHA::TRP1* |
| Su e-3xHA | *MATa; ura3-52, lys2-801amber} ade2-101ocre, trp1-Δ63, his3-Δ200, leu2-Δ1} or BY4724 {MATα; his3Δ1; leu2Δ0; lys2Δ0; ura3Δ0; Su e-3xHA::HIS3* |
| Δmgm1 | *MATa; ura3-52, lys2-801amber, ade2-101ocre, trp1-Δ63, his3-Δ200, leu2-Δ1, mgm1::KAN* |
| Mgm1-3xHA | *MATa; ura3-52, lys2-801amber, ade2-101ocre, trp1-Δ63, his3-Δ200, leu2-Δ1, MGM1-3xHA::TRP1* |
| Δmic60 F1β-3xHA | *MATa; ura3-52, lys2-801amber, ade2-101ocre, trp1-Δ63, his3-Δ200, leu2-Δ1, mic60::His3, F1β-3xHA::TRP1* |
| Cox5A-3xHA | *MATa; ura3-52, lys2-801amber, ade2-101ocre, trp1-Δ63, his3-Δ200, leu2-Δ1} or BY4724 {MATα; his3Δ1; leu2Δ0; lys2Δ0; ura3Δ0; COX5A-3xHA::KAN* |
| Δmic60 Cox5A-3xHA | *MATa; ura3-52, lys2-801amber, ade2-101ocre, trp1-Δ63, his3-Δ200, leu2-Δ1, mic60::His3, COX5A-3xHA::KAN* |
| Δdnm1 | *MATa; ura3-52, lys2-801amber, ade2-101ocre, trp1-Δ63, his3-Δ200, leu2-Δ1, dnm1::KAN* |
| Δmic10 | *MATa; ura3-52, lys2-801amber, ade2-101ocre, trp1-Δ63, his3-Δ200, leu2-Δ1} or BY4724 {MATα; his3Δ1; leu2Δ0; lys2Δ0; ura3Δ0; mic10::KAN* |
| Δmic60 | *MATa; ura3-52, lys2-801amber, ade2-101ocre, trp1-Δ63, his3-Δ200, leu2-Δ1} or BY4724 {MATα; his3Δ1; leu2Δ0; lys2Δ0; ura3Δ0; mic60::HIS3* |
| Δdnm1 Δmgm1 | *MATa; ura3-52, lys2-801amber, ade2-101ocre, trp1-Δ63, his3-Δ200, leu2-Δ1, dnm1::KAN, mgm1::hphNT1* |
| Δdnm1 Δfzo1 | *MATa, ura3-52, lys2-801amber, ade2-101ocre, trp1-Δ63, his3-Δ200, leu2-Δ1, dnm1::HIS3, fzo1::KAN, rho+* |
| Δdnm1 Δmic10 | *MATa; ura3-52, lys2-801amber, ade2-101ocre, trp1-Δ63, his3-Δ200, leu2-Δ1, dnm1::KAN, mic10::hphNT1* |
| Δdnm1 Δmic60 | *MATa; ura3-52, lys2-801amber, ade2-101ocre, trp1-Δ63, his3-Δ200, leu2-Δ1, dnm1::KAN, mic60::HIS3* |
| Δdnm1 Δsu e | *MATa; ura3-52, lys2-801amber, ade2-101ocre, trp1-Δ63, his3-Δ200, leu2-Δ1, dnm1::KAN, su e::HIS3* |
| Δdnm1 Δmgm1 Δmic60 | *MATa; ura3-52, lys2-801amber, ade2-101ocre, trp1-Δ63, his3-Δ200, leu2-Δ1} or BY4724 {MATα; his3Δ1; leu2Δ0; lys2Δ0; ura3Δ0; dnm1::KAN, mic60::HIS3; mgm1::hphNT1* |
| Δdnm1 Δsu e Δmic60 | *MATa; ura3-52, lys2-801amber, ade2-101ocre, trp1-Δ63, his3-Δ200, leu2-Δ1} or BY4724 {MATα; his3Δ1; leu2Δ0; lys2Δ0; ura3Δ0; dnm1::KAN, mic60::HIS3; su e::HIS3* |

# Materials and methods

## Yeast strains and growth conditions

*S. cerevisiae* strains YPH499 or W303 were used as wild type (WT). Genetic manipulations were performed according to standard procedures (*Longtine et al., 1998*). The genotypes are listed in *Table 1*. Cells were grown as indicated on YPD (1% yeast extract, 2% peptone, 2% glucose), YPGal (1% yeast extract, 2% peptone, 2% galactose) or YPG (1% yeast extract, 2% peptone, 3% glycerol) (*Sherman, 1991*). *mgm1-5* cells were kept at 25°C to avoid loss of mtDNA. Loss of Mgm1 function was induced by resuspension of cells in pre-warmed liquid medium and subsequent incubation at 37°C for 25 min. For restoration of Mgm1 function, cells were resuspended in liquid medium (25°C) and incubated for additional 30 min.

## Isolation of mitochondria

Mitochondria were isolated as described (*Lee et al., 1998*). Cells were cultured to a final $OD_{600}$ of 0.8–1.2 and harvested by centrifugation for 5 min at 2000 xg at RT. Cells were washed with 40 ml water and the wet weight was measured. They were resuspended in 30 ml of alkaline solution (100 mM Tris (pH not adjusted), 10 mM DTT) and incubated for 30 min at 30°C under gentle agitation. Cells were harvested, washed once with spheroplast buffer (20 mM Tris-HCl pH 7.4, 1 mM EDTA,

1.2 M sorbitol), resuspended in spheroplast buffer (10 ml/g wet weight) containing 6.6 mg Zymolyase per g wet weight and incubated for 30 min at 30°C. Spheroplasts were harvested by centrifugation at 2000 xg at 4 °C for 5 min and washed twice with 15 ml ice-cold lysis buffer (20 mM MOPS-KOH pH 7.2, 1 mM EDTA, 0.6 M sorbitol, 0.2 % (w/v) BSA, 1 mM PMSF). Spheroplasts were resuspended in 15 ml ice-cold lysis buffer by pipetting using a Pipetman P5000 (Gilson, Middleton, USA) with 1cm cutoff P5000 tips. Unbroken spheroplasts were harvested by centrifugation at 2000 xg at 4 °C for 5 min and again resuspended in ice-cold lysis buffer. The supernatants were collected, pooled and mitochondria were harvested by centrifugation at 12,000 xg and 4°C for 5 min. Mitochondria were washed once with SM buffer (20 mM MOPS, pH 7.4, 0.6 M sorbitol) and finally resuspended in SM buffer.

## Subfractionation of mitochondria

Generation and separation of membrane vesicles were performed as described previously (*Harner et al., 2011*). 10 mg of freshly isolated mitochondria were resuspended in SM buffer, then 20 ml swelling buffer (20 mM MOPS, pH 7,4, 0.5 mM EDTA, 1 mM PMSF, 1x Roche complete protease inhibitor) were added dropwise, followed by incubation under continuous stirring for 30 min on ice. 5 ml 2.5 M sucrose were added and the suspension was incubated for 15 min on ice. The suspension was subjected to sonication for 3 times 30 s with 30 s breaks on ice at 60% duty cycle and output control 0 (Branson Sonifier 250; Branson, Danbury, USA) using a microtip. Residual intact mitochondria were removed by centrifugation for 20 min at 20,000 xg and 4°C. The submitochondrial vesicles were concentrated on a 2.5 M sucrose cushion by high speed centrifugation (120,000 xg for 100 min at 4°C). The vesicle pellet was resuspended, loaded under a continuous sucrose gradient (0.8-1.25 M sucrose in 20 mM MOPS, pH 7.4, 0.5 mM EDTA) and separated by centrifugation for 24 h at 200,000 xg and 4°C. The gradient was fractionated in 500 µl fractions, the proteins were TCA precipitated and analyzed by SDS-PAGE and immunoblotting.

## Drop dilution assay

Cells were grown in YPG liquid medium and kept in logarithmic phase, washed once with water, diluted in water to an $OD_{600}$ of 0.3. Afterwards serial dilutions were performed (1:10; 1:100; 1:1000; 1:10,000). 3 µl of each dilution were spotted on agar plates containing the indicated media and were incubated at 30°C.

## Maintenance of mtDNA and DAPI staining

Yeast strains were grown on YPG plates to select for mtDNA, transferred to liquid YPD medium to allow for loss of mtDNA and continuously kept in the logarithmic growth phase. At time point 0 and after 48 hr the percentage of cells containing mtDNA was determined by DAPI staining. DAPI staining was performed as described (*Harner et al., 2014*).

## Generation of yeast cell extract

For analysis of cell extracts the strains were grown to logarithmic phase. Cells ($OD_{600}$ 12.5) were harvested by centrifugation, washed with water and resuspended 1ml water containing 100 µg PMSF. Cells were lysed by incubation for 10 min in 0.25 M NaOH and 1 % β-mercapto ethanol (final concentration). Proteins were subjected to TCA precipitation and acetone washed. The pellets were resuspended in 100 µl Laemmli buffer.

## Electron microscopy

Electron microscopy of chemically fixed cells was performed as described (*Bauer et al., 2001*). Ultrathin sections were collected on Pioloform-coated copper slot grids (Plano, Wetzlar, Germany) and stained with uranyl acetate and lead citrate (*Reynolds, 1963*). For electron microscopy after cryosectioning (Tokuyasu method), cells were grown to exponential phase in YPD or YPG medium, chemically fixed, embedded in 12% gelatin and cryo-sectioned as described previously (*Griffith et al., 2008*). Ultrathin cryo-sections were collected with a 1:1 mixture of 2% methylcellulose (in ddH2O) and 2.3 M, sucrose 120 mM PIPES, 50 mM HEPES, pH 6.9, 4 mM $MgCl_2$, 20 mM EGTA and layered on Formvar/carbon coated copper 100 mesh grids. Structural phenotypes were analyzed by evaluation of 100 mitochondrial profiles unless otherwise indicated. Immunological reactions were

performed with monoclonal anti-HA (RRID:AB_514505) and a protein A-gold 10 nm conjugate. Protein localization gold particles present in 100 cells were counted. Samples for the electron tomography analyses were processed as previously described (*Mari et al., 2014*).

### Blue native gel electrophoresis (BN-PAGE)

75 µg of mitochondria were pelleted by centrifugation and resuspended in BN-lysis buffer (*Wittig et al., 2006*). For analysis of $F_1F_O$-ATP synthase or MICOS complex 1 % (w/v) or 3 % (w/v) digitonin were used. Cleared lysates were supplemented with Native PAGE 5% G-250 Sample Additive and subjected to BN-PAGE (Native PAGE 3–12% Bis-Tris; Life Technologies, Carlsbad, CA, USA). After blotting on PVDF membranes (Roth, Karlsruhe, Germany) immuno-decoration using the indicated antibodies was performed.

## Acknowledgements

We thank Dr. Jodi Nunnari for providing us with the *mgm1-5* mutant, Matthew Harmey for discussion and Monika Krause, MPI for Biochemistry, for the artwork. WN is grateful to the Max Planck Society for a Senior Fellowship and the Carl Friedrich von Siemens Stiftung for financial support. MEH and WN acknowledge the generous support and advice by Dr. Michael Kiebler, BMC, University of Munich. MEH thanks the Jung-Stiftung für Wissenschaft und Forschung and LMUexcellent for financial support. FR is supported by ALW Open Program (822.02.014), DFG-NWO cooperation (DN82-303), SNF Sinergia (CRSII3_154421) and ZonMW VICI (016.130.606) grants. A-KU thanks the Max Planck Society for a Ph.D. Fellowship and acknowledges the support by the Elite Network Bavaria, Macromolecular Science.

## Additional information

### Funding

| Funder | Grant reference number | Author |
|---|---|---|
| Max-Planck-Gesellschaft | | Max E Harner<br>Ann-Katrin Unger<br>Toshiaki Izawa<br>Walter Neupert |
| Carl Friedrich von Siemens Stiftung | | Walter Neupert |
| Jung-Stiftung für Wissenschaft und Forschung | | Max E Harner |
| Ludwig-Maximilians-Universität München | | Max E Harner |
| Netherlands organization for Scientific Research | DN82-303 | Fulvio Reggiori |
| Deutsche Forschungsgemeinschaft | DN82-303 | Fulvio Reggiori |
| Schweizerischer Nationalfonds zur Förderung der Wissenschaftlichen Forschung | CRSII3_154421 | Fulvio Reggiori |
| ZonMw | ZonMW VICI | Fulvio Reggiori |
| Netherlands organization for Scientific Research | 822.02.014 | Fulvio Reggiori |

The funders had no role in study design, data collection and interpretation, or the decision to submit the work for publication.

### Author contributions

MEH, Conception and design, Acquisition of data, Analysis and interpretation of data, Drafting or revising the article; A-KU, WJCG, MM, TI, MS, Acquisition of data, Analysis and interpretation of

data; SG, FR, Analysis and interpretation of data, Drafting or revising the article; BW, WN, Conception and design, Analysis and interpretation of data, Drafting or revising the article

## Author ORCIDs
Benedikt Westermann, http://orcid.org/0000-0002-2991-1604
Walter Neupert, http://orcid.org/0000-0003-0571-4419

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
