## [Decision Letter]

Thank you for submitting your article "Two pathways of crista formation determine mitochondrial architecture and homeostasis" for consideration by *eLife*. Your article has been favorably evaluated by Randy Schekman (Senior Editor) and three reviewers, one of whom is a member of our Board of Reviewing Editors. The reviewers have opted to remain anonymous.

The reviewers have discussed the reviews with one another and the Reviewing Editor has drafted this decision to help you prepare a revised submission.

Summary:

This is a highly intriguing manuscript that reports a new model of formation of the crista structures in yeast mitochondria. Mitochondria are cellular organelles with a characteristic shape and inner architecture. The most fascinating ultrastructural feature of mitochondria is the presence of numerous, heterogeneously shaped invaginations of the inner membrane that are the main sites of oxidative phosphorylation in eukaryotic cells and have been termed cristae membranes. Cristae are considered to represent a specialized micro-compartment within mitochondria that is ideally suited to support ATP synthesis by chemiosmotic coupling. Surprisingly, very little is known about the mechanisms of cristae biogenesis. It has been shown that dimerization and oligomerization of ATP synthase complexes and the MICOS complex that connects cristae membranes to the peripheral inner membrane regions are critical for normal cristae morphology and function. But a comprehensive and conclusive model how differently shaped cristae are formed has been lacking.

Harner et al. developed a very elegant hypothesis to explain the formation of different types of cristae. The authors classified the cristae into lamellar cristae and tubular cristae and then analyzed those different crista structures in various yeast deletion and temperature-sensitive mutants by EM. They found that lamellar cristae are formed by Mgm1, which mediates inner membrane fusion, in cooperation with the MICOS complex, the crista junction forming machinery, and the ATPase dimer, which forms highly bended crista rims. On the other hand, they found that tubular cristae are formed by the MICOS complex and ATPase dimer, but not by Mgm1. They proposed a new model on how the IM fusion by Mgm1 leads to lamellar cristae together with the MICOS complex and ATPase dimer. This overall conclusive hypothesis is consistent with earlier studies on mitochondrial morphology and dynamics and several key points are supported by new experimental data. The cornerstones of this model are: 1) Tubular and lamellar cristae are formed via different mechanisms. 2) Both pathways require the dimerization of ATP synthase and the MICOS machinery. 3) Tubular cristae are formed by triggering inner membrane invaginations, probably at MICOS complexes, that grow through vectorial transport of proteins and lipids into the emerging tubule. 4) Formation of lamellar cristae additionally requires the inner membrane fusion activity of Mgm1/OPA1. 5) The inner membrane fusion activity of Mgm1 is limited to sites of mitochondrial outer and inner membrane contacts. 6) Therefore, inner membrane fusion events leave behind double membrane sheets derived from the original sites of inner membrane tethering that finally become lamellar cristae. 7) Membrane bending at the rims of these new-born lamellar cristae is stabilized by dimerization and oligomerization of ATP synthase complexes. 8) Mgm1-mediated inner membrane fusion is stopped at the position of inner membrane MICOS complexes leading to the formation of crista junctions. In this way the MICOS complex prevents the complete detachment of the inner membrane fusion septum from the peripheral inner membrane.

The experiments were conducted carefully and the experimental quality is very high. The findings reported in this manuscript provide an important basis for future studies on the mechanism of formation and regulation of the complex mitochondrial membrane architecture.

Essential revisions:

1) The data presented in this study clearly support: (i) the direct involvement of Mgm1 activity in cristae formation; (ii) the origin of the internal membrane stacks observed in MICOS-deficient mutants from cristae-like structures, and (iii) the strong prevalence of tubular cristae in mitochondria with reduced Mgm1 activity, like in Dnm1-deficient cells. These are important new findings that were possible mainly by the elegant combination of yeast mutants and advanced electron cryo-tomography technology. Although very thoughtful, elegant and consistent with the available literature, the further proposals on the mechanisms of cristae formation remain hypothetical. The hypothetical nature of the suggested model should be emphasized in the paper, including title and abstract. The manuscript should be rewritten to clearly state that the authors present a novel hypothesis that is based in important parts on the data shown, but you may also choose to point out which future studies will be interesting to address various aspects of the hypothesis.

2) The finding that the loss of cristae upon transient inactivation of Mgm1 can be reversed rapidly (within a time period of 25 min) is novel and very interesting. What is the possible mechanism of such a quick disappearance of cristae without degradation of respiratory chain complexes or ATPases upon shift to 37°C? Similarly, Mgm1-mediated crista re-formation should depend on mitochondrial fusion, but such fusion may not be that frequent after temperature down-shift to 25°C. An explanation of this point is required.

Since 25 min is much less than one generation time in yeast, this result implies a very high turnover of cristae membranes also in healthy cells. Is there evidence for such a phenomenon? Since the respiratory competence of the cells is maintained, what happens to the respiratory chain complexes when cristae disappear? What is the morphology of the remaining cristae in the mgm1 conditional mutant after the short heat shock? Finally, when the mgm1 conditional mutant is shifted back to 25°C after the short heat shock, cristae re-appear. Is the overall morphology of the mitochondrial network also restored?

It should be discussed if Mgm1 may affect inner membrane morphology independently of its role in inner membrane fusion. A novel function of Mgm1 in cristae biogenesis may also proceed via inner membrane fusion in the absence of outer membrane fusion; thus, lamellar cristae may arise via Mgm1-mediated lateral fusion of tubular cristae. Is there a requirement for outer membrane fusion in the generation of lamellar cristae? What is known about cristae architecture in a double deletion of Dnm1 and Fzo1?

3) Since the observation of crista structures requires EM analyses, it is difficult to follow the process of cristae formation at a high time-resolution. Therefore most of the data except for Figure 1 are the consequences of steady-state depletion of the components (Dnm1, Mic60, Su e etc.) involved in mitochondrial structural dynamics, with the risk of possible secondary effects. On the other hand, the authors showed that cristae formation and disappearance are rather rapid processes that takes place within 25 min in Figure 1. Thus, temperature-sensitive mutants or promoter shut-off of those components could be used to minimize the possibility of secondary effects arising from the steady-state depletion. This may be beyond the scope of the present work, but these points should be discussed and where possible complemented by experimental findings.

4) The authors argue that dimeric ATP synthase, which generates membrane curvature, as well as MICOS, which stabilizes crista junctions, are required for the generation of both types of cristae. It should be discussed that MICOS mutants lack crista junctions, yet do contain cristae-like lamellar inner membrane compartments, as the authors also demonstrate. Moreover, they show that dnm1-δ mic60-δ mitochondria contain tubular as well as lamellar internal membranes. Thus, MICOS or at least Mic60 may not be strictly required for cristae biogenesis, but for crista junction maintenance.

ATP synthase dimers are evidently required for the biogenesis of tubular cristae. The authors suggest that septated inner membranes may arise upon fusion of the outer membrane that is not followed by inner membrane fusion due to lack of membrane curvature. This hypothesis is exciting. A su e-δ fzo1-ts strain would be interesting here. The authors report that the levels of assembled MICOS complexes are reduced in the absence of Su e and thus ATP synthase dimerization, but increased in the absence of Dnm1. Many different forms and subcomplexes of MICOS have been suggested, which raises the question, what is the nature of the molecular species detected with Mic27 antibodies in native gels western blots? Does MICOS dissociate into the recently demonstrated Mic10- and Mic60-containing subcomplexes in Su e-deficient mitochondria? Why does Dnm1 depletion enhance the level of the assembled MICOS complex? The authors reasoned that the decreased Dnm1 activity results in reduced formation of lamellar cristae, but why do tubular cristae structures increase?

[Editors' note: further revisions were requested prior to acceptance, as described below.]

Thank you for resubmitting your work entitled "An evidence-based hypothesis on the homeostasis of mitochondrial architecture" for further consideration at *eLife*. Your article has been favorably evaluated by Randy Schekman (Senior Editor) and three reviewers, one of whom, Klaus Pfanner, is a member of our Board of Reviewing Editors.

The article has been carefully revised and includes substantial additional data. It presents an innovative and elegant hypothesis on the formation and organization of mitochondrial cristae and provides important evidence to support the hypothesis.

The reviewers raised one final point:

The new title is good in pointing out the nature of the work as a hypothesis, but sounds a bit general. You may consider to include part of the previous title (i.e. " two pathways of crista formation") in the new title.

---

## [Author Response]

[…]

Essential revisions:

1) The data presented in this study clearly support: (i) the direct involvement of Mgm1 activity in cristae formation; (ii) the origin of the internal membrane stacks observed in MICOS-deficient mutants from cristae-like structures, and (iii) the strong prevalence of tubular cristae in mitochondria with reduced Mgm1 activity, like in Dnm1-deficient cells. These are important new findings that were possible mainly by the elegant combination of yeast mutants and advanced electron cryo-tomography technology. Although very thoughtful, elegant and consistent with the available literature, the further proposals on the mechanisms of cristae formation remain hypothetical. The hypothetical nature of the suggested model should be emphasized in the paper, including title and abstract. The manuscript should be rewritten to clearly state that the authors present a novel hypothesis that is based in important parts on the data shown, but you may also choose to point out which future studies will be interesting to address various aspects of the hypothesis.

According to the advice by the referees we have rewritten Title, Abstract, Introduction, Results and Discussion to point out that we present a hypothesis based on our results. We also have added several sentences to the end of the Discussion to point out at least a few of the numerous new aspects that are opened by our study.

2) The finding that the loss of cristae upon transient inactivation of Mgm1 can be reversed rapidly (within a time period of 25 min) is novel and very interesting. What is the possible mechanism of such a quick disappearance of cristae without degradation of respiratory chain complexes or ATPases upon shift to 37°C? Similarly, Mgm1-mediated crista re-formation should depend on mitochondrial fusion, but such fusion may not be that frequent after temperature down-shift to 25°C. An explanation of this point is required.

The rate of mitochondrial fusion in yeast, as defined by the number of fusion events per min and cell, has been determined to be about 1 event or higher. The fusion events measured in this way comprise quite a number of steps including establishing contact with another mitochondrion, fusion of the outer membrane, all complex reactions. The situation in the experiment addressed here is very different since the OM is fused and the reactivated Mgm1 is ready for immediately starting fusion of two aligned membranes. In fact, the experiment suggested by one of the referees, the analysis of the mitochondrial morphology in *mgm1-5* cells upon temperature shifts, points also in this direction (see also further down). Cristae reappear faster than the mitochondrial network. This indicates, that reactivated Mgm1 does use the septa formed at non-permissive temperature to make new cristae. Therefore, in this special situation the generation of cristae seems to be independent of an outer membrane fusion event. We have now explained this in more detail in the Discussion section (third paragraph).

It would certainly be very interesting to try to measure the various step times, but this is practically impossible in view of the lack of the necessary tools. However, we have commented on this in the revised manuscript to clarify the situation.

Since 25 min is much less than one generation time in yeast, this result implies a very high turnover of cristae membranes also in healthy cells. Is there evidence for such a phenomenon? Since the respiratory competence of the cells is maintained, what happens to the respiratory chain complexes when cristae disappear? What is the morphology of the remaining cristae in the mgm1 conditional mutant after the short heat shock? Finally, when the mgm1 conditional mutant is shifted back to 25°C after the short heat shock, cristae re-appear. Is the overall morphology of the mitochondrial network also restored?

A) We have analyzed the assembly state of Complex III of the respiratory chain by BNGE. Complex III of *mgm1-5* incubated at 37°C cannot be distinguished from that of the mutant grown at 25°C nor from that of wild type. Please find the respective data in the text (subsection “Mgm1 plays a direct role in cristae formation”, second paragraph) and in Figure 2—figure supplement 4.

B) We performed a tomographic analysis and most interestingly found that the remaining cristae in the *mgm1-5* mutant after exposure to 37°C are tubular. This further strongly supports our hypothesis that lamellar cristae are formed by Mgm1 mediated inner membrane fusion. These data are now added to the manuscript (subsection “Mgm1 plays a direct role in cristae formation”, fourth paragraph and Figure 2). In addition, we provide two further videos (Video 2).

C) We determined the morphology of the mitochondria during the whole temperature shift experiment showing fragmentation and reformation of the mitochondrial network after return to 25°C. We provide these data in the third paragraph of subsection “Mgm1 plays a direct role in cristae formation” and in the new Figure 2.

It should be discussed if Mgm1 may affect inner membrane morphology independently of its role in inner membrane fusion. A novel function of Mgm1 in cristae biogenesis may also proceed via inner membrane fusion in the absence of outer membrane fusion; thus, lamellar cristae may arise via Mgm1-mediated lateral fusion of tubular cristae. Is there a requirement for outer membrane fusion in the generation of lamellar cristae? What is known about cristae architecture in a double deletion of Dnm1 and Fzo1?

A) The referee is right in pointing out that one cannot rule out an additional role of Mgm1. We also do not rule out that lamellar or tubular cristae can fuse to a certain extent. We would, however, exclude a role of Mgm1 in lateral fusion of tubular cristae to form lamellar cristae. This is because all major domains of Mgm1 that are required for fusion activity are located on the inside of tubular cristae and therefore would have the wrong topology for being able to catalyze fusion.

B) The shift and backshift experiments with *mgm1-5* show that fusion of septa occurs when the outer membrane is fused. In the situation when two mitochondria fuse, initially the outer membrane must fuse.

C) We have generated the △*dnm1* △*fzo1* double deletion strain and analyzed the cristae in this mutant. These experiments are now included in the Results section (subsection “Mitochondrial fusion/fission dynamics contributes to the formation of lamellar cristae”, second paragraph; and Figure 7—figure supplement 1). In short, they show that also in this mutant tubular crista are formed, but not to the same extent as in the △*dnm1* △*mgm1* double mutant, which is explained by Dnm1 independent IM fission in the absence of Fzo1.

3) Since the observation of crista structures requires EM analyses, it is difficult to follow the process of cristae formation at a high time-resolution. Therefore most of the data except for Figure 1 are the consequences of steady-state depletion of the components (Dnm1, Mic60, Su e etc.) involved in mitochondrial structural dynamics, with the risk of possible secondary effects. On the other hand, the authors showed that cristae formation and disappearance are rather rapid processes that takes place within 25 min in Figure 1. Thus, temperature-sensitive mutants or promoter shut-off of those components could be used to minimize the possibility of secondary effects arising from the steady-state depletion. This may be beyond the scope of the present work, but these points should be discussed and where possible complemented by experimental findings.

Yes, we agree, steady state depletion of components is not without the risk of secondary effects. We agree also that more ts-mutants would be desirable. Promoter shutoff would certainly also be useful. We will certainly aim for obtaining more ts-mutants in particular mutants of subunits of F_1_F_O_–ATP synthase required for dimerization and of MICOS components. On the other hand, such mutants are often leaky and can pose problems of other kinds.

*4) The authors argue that dimeric ATP synthase, which generates membrane curvature, as well as MICOS, which stabilizes crista junctions, are required for the generation of both types of cristae. It should be discussed that MICOS mutants lack crista junctions, yet do contain cristae-like lamellar inner membrane compartments, as the authors also demonstrate. Moreover, they show that dnm1-δ mic60-δ mitochondria contain tubular as well as lamellar internal membranes. Thus, MICOS or at least Mic60 may not be strictly required for cristae biogenesis, but for crista junction maintenance.*

A) This is an important and interesting point. Indeed, the internal membranes in the Mic60 and Mic10 deletion mutants are very much like normal cristae. We consider it as strength of our hypothesis that we can explain for the first time, how these membranes are made and that they are able to perform OXPHOS and other crista functions. According to the advice of the referee we have extended the Discussion to make this issue very clear (fifth paragraph).

B) The tubular membrane structures in dnm1-δ mic60-δ mitochondria that accumulate and have the tendency to fuse or break into vesicles are quite weird. We explain this by the lack of crista junctions which do not form regular tubular cristae but simply deliver the incoming proteins and lipids into the mitochondrial interior, because the inner boundary membrane cannot be expanded. We believe that such membranes must have a tendency to fuse like lipid vesicles in contrast to the well-organized normal membrane structures. Indeed, these cells grow extremely poorly, lose mitochondrial DNA and die rapidly. We believe that they cannot be used to argue about the behavior of normal mitochondrial membranes.

ATP synthase dimers are evidently required for the biogenesis of tubular cristae. The authors suggest that septated inner membranes may arise upon fusion of the outer membrane that is not followed by inner membrane fusion due to lack of membrane curvature. This hypothesis is exciting. A su e-δ fzo1-ts strain would be interesting here. The authors report that the levels of assembled MICOS complexes are reduced in the absence of Su e and thus ATP synthase dimerization, but increased in the absence of Dnm1. Many different forms and subcomplexes of MICOS have been suggested, which raises the question, what is the nature of the molecular species detected with Mic27 antibodies in native gels western blots? Does MICOS dissociate into the recently demonstrated Mic10- and Mic60-containing subcomplexes in Su e-deficient mitochondria? Why does Dnm1 depletion enhance the level of the assembled MICOS complex? The authors reasoned that the decreased Dnm1 activity results in reduced formation of lamellar cristae, but why do tubular cristae structures increase?

A) Su e is epistatic over the other key components discussed here. Deletion of Su e always leads to mitochondria with septa or onion-like mitochondria. Therefore, we have not made this mutant.

B) We have no experimental evidence that MICOS dissociates into distinct subcomplexes. However, we cannot rule this out. It might be possible that these subcomplexes are not detectable with our antibodies. We analyzed the molecular composition of the Mic27 containing complex in the Su e deletion strain and found no difference as compared to WT. The question about the role of MICOS subcomplexes in MICOS assembly is certainly very interesting question and it would be worthwhile to follow it in a detailed investigation in view of the results of Bohnert et al., 2015; Friedman et al., 2015; Zerbes et al., 2016. We believe that this would require a separate extensive study.

C) We observe that the number of crista junctions increase markedly in the *∆dnm1* and in the *∆dnm1∆mgm1* deletion mutants. We interpret this by the necessity of these mitochondria to accommodate their OXPHOS and other components. The surface area generated by lamellar cristae is clearly much higher than by a tubule when related to a crista junction. We are mentioning this now in the text (subsection “Mitochondria lacking Dnm1 show altered crista structure”).